# Single-molecule imaging correlates decreasing nuclear volume with increasing TF-chromatin associations during zebrafish development

Matthias Reisser[1], Anja Palmer[1], Achim P. Popp[1], Christopher Jahn[2], Gilbert Weidinger [2] & J. Christof M. Gebhardt[1]

Zygotic genome activation (ZGA), the onset of transcription after initial quiescence, is a major developmental step in many species, which occurs after ten cell divisions in zebrafish embryos. How transcription factor (TF)-chromatin interactions evolve during early development to support ZGA is largely unknown. We establish single molecule tracking in live developing zebrafish embryos using reflected light-sheet microscopy to visualize two fluorescently labeled TF species, mEos2-TBP and mEos2-Sox19b. We further develop a data acquisition and analysis scheme to extract quantitative information on binding kinetics and bound fractions during fast cell cycles. The chromatin-bound fraction of both TFs increases during early development, as expected from a physical model of TF-chromatin interactions including a decreasing nuclear volume and increasing DNA accessibility. For Sox19b, data suggests the increase is mainly due to the shrinking nucleus. Our single molecule approach provides quantitative insight into changes of TF-chromatin associations during the developmental period embracing ZGA.

[1] Institute of Biophysics, Ulm University, Albert-Einstein-Allee 11, 89081 Ulm, Germany. [2] Institute of Biochemistry and Molecular Biology, Ulm University, Albert-Einstein-Allee 11, 89081 Ulm, Germany. Correspondence and requests for materials should be addressed to J.C.M.G. (email: christof.gebhardt@uni-ulm.de)

Zebrafish embryos undergo several major morphogenetic transitions during early development, including the mid-blastula transition (MBT) at the 1000-cell stage and gastrulation at dome stage[1]. During MBT, the embryo dramatically switches its transcription program in the maternal-to-zygotic transition, as maternally inherited mRNA is degraded and the zygotic genome is activated (zygotic genome activation, ZGA) (Fig. 1a), accompanied by an increase in cell cycle length[1,2]. Until gastrulation, the volume of the animal cap is approximately constant, while individual cells decrease in size after each cell division, accompanied by a decreasing nucleus, but increasing nucleo-cytoplasmic volume ratio[1,3].

ZGA is well characterized on the level of mRNA transcripts, for example the relative occurrence of maternal and zygotic transcript levels at different developmental stages[4,5]. In contrast, much less is known about how chromatin binding of proteins initiating transcription such as transcription factors (TFs) or components of the transcription machinery changes during early development. In particular, it is unclear whether TFs are able to bind to DNA in early developmental stages and how the DNA-bound fraction or kinetic properties of TFs change during development.

Several physical and biological factors are expected to influence TF binding to chromatin. In a framework assuming equilibrium of TF-chromatin interactions, TF-chromatin binding is governed by the law of mass action (Fig. 1b). Increasing TF levels will increase the absolute number of chromatin-bound TF molecules in non-saturating conditions. In addition, also the percentage of chromatin-bound TF molecules, i.e., the chromatin-bound fraction of a TF, might undergo changes, as it depends on the concentration of TF binding sites and on the rate constants of association and dissociation of the TF (Fig. 1b). Binding site concentration in turn is determined by dividing the number of binding sites accessible to this TF on chromatin by the volume of the nucleus. In general, any physical or biological factor will either directly or indirectly affect one of the parameters nuclear size, kinetic rate constants or number of accessible binding sites. In particular, non-equilibrium processes typically observed in biological systems will also be reflected in changes of these parameters.

Nuclear volume decreases during early zebrafish development[1]. The overall amount of accessible DNA has been shown to increase during development in Drosophila and human embryos, possibly due to a combined action of pioneering TFs and chromatin remodelers[6,7]. The apparent number of accessible binding sites on chromatin might also increase during development due to titration of a binding repressor via an increase in nucleo-cytoplasmic volume ratio[8], resulting in a decrease in binding competition between TF and histones[3,9]. Changes in cofactor abundance[10] that might present additional binding sites might have an additional effect on the apparent number of accessible binding sites. Taken together, these changes in nuclear volume and DNA accessibility are expected to increase the chromatin-bound fraction of TF molecules.

Kinetic rate constants might also change during development, for example due to changes in the presence of binding cofactors stabilizing TF-chromatin interactions[10]. Additionally, it has been shown that some TFs displace each other from DNA at high TF concentrations, thereby destabilizing TF-chromatin interactions, an effect termed facilitated dissociation[11]. Thus, changes in TF-chromatin binding kinetics might further alter the chromatin-bound fraction of a TF.

Binding of TFs to DNA and the assembly of the transcriptional machinery are intrinsically stochastic processes[12–14]. Thus, single molecule imaging has evolved as the method of choice to obtain kinetic and quantitative information on TF-DNA interactions.

Single molecule imaging has been performed in bacteria[15,16], individual eukaryotic cells[17–21], large salivary gland cells, or cell spheroids[22–24] and whole embryos or adult organisms[25–28]. In living zebrafish, single molecule imaging of YFP-tagged membrane proteins has been performed in epithelial cells at the surface of live zebrafish using total internal reflection fluorescence (TIRF) microscopy[25].

Examples of TFs active during early zebrafish development are the general TF TATA-binding protein (TBP) and the TF Sox19b. TBP is a member of the transcription complex and important for the assembly of general TFs to form the pre-initiation complex[29]. It is highly translated in early development[30] and necessary for transcription of many genes in zebrafish[31]. Sox19b is functionally related to Sox2[32] and an important transcription activator in early zebrafish development[33,34].

Here, we establish single molecule imaging deep within live growing zebrafish embryos to study chromatin binding of mEos2-TBP and mEos2-Sox19b during the time course of development embracing ZGA. We find that both TF species are able to bind to chromatin as early as the 64-cell stage. For both TF species, we monitor the chromatin-bound fraction and observe an increase in the chromatin-bound fraction during this time period. In the case of TBP, the increase is compatible with a model of TF-chromatin interactions including a decreasing nuclear volume and an increase in the apparent number of chromatin binding sites. For Sox19b, our data suggest that increasing chromatin associations are mainly due to the decreasing nuclear volume alone. Our quantitative single molecule data raise the possibility that the decreasing nuclear volume facilitates ZGA in zebrafish embryos by enhancing TF-chromatin associations.

## Results

**Single molecule imaging in live zebrafish embryos by RLSM.** To image individual TF molecules in the nucleus of zebrafish cells, we adapted reflected light-sheet microscopy (RLSM), originally developed for single fluorescent protein imaging in live mammalian cells[21], to the specific requirements given by the size, shape and medium conditions of live zebrafish embryos (Fig. 2a, Supplementary Figure 1 and Methods). In particular, we used a 1.5 mm × 3 mm large mirror to reflect a ca. 3 μm thick sheet of light created by a vertically mounted illumination objective into a horizontal plane. This plane was aligned to coincide with the focal plane of a high numerical aperture detection objective. The mirror could be positioned by remote control, which allowed for user-friendly alignment and handling of the RLS microscope. Both illumination objective and mirror were mounted onto a rotary stage to enable illumination of embryos from various angles. We positioned the embryo horizontally using a piezo xy-stage and scanned it vertically using a long-range piezo z-stage.

TBP and Sox19b of zebrafish origin were fused to the fluorescent protein mEos2, analogous to previous fusion proteins of TBP and a Sox family member, Sox2[35–37] (Fig. 2b, Supplementary Table 1 and 4 and Methods). It has been shown that a Halo-tag knocked in to the N-terminus does not compromise TBP and Sox2 function[36,37]. We microinjected mRNA coding for either mEos2-TF fusion protein into dechorionated fertilized eggs at the 1-cell stage (Fig. 2b, and see Methods). To avoid saturation of binding sites, we injected low amounts of mRNA. 60 pg of mRNA increased ectopic mEos2-TBP expression at the 1000 cell stage 3.8 ± 2.5 (mean ± s.d.) fold over endogenous TBP levels, as shown by Western Blot (Fig. 2c, Supplementary Figure 2 and see Methods). Both endogenous TBP and ectopic mEos2-TBP abundance increased from 64-cell to oblong stage. For Sox19b, we decreased injected mRNA further to 29 pg. With these amounts of mRNA, neither mEos2-TBP nor mEos2-Sox19b

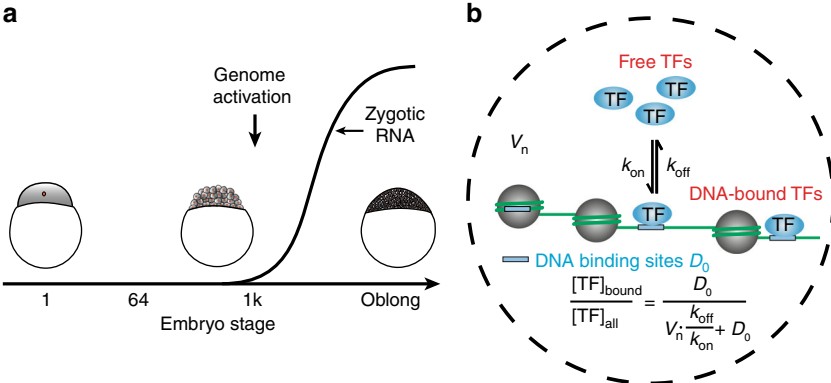

**Fig. 1** TF binding and zygotic genome activation. **a** Sketch of zygotic mRNA levels as a function of embryo stage. Inset: schemes of embryos at different developmental stages. **b** Scheme of TF binding to chromatin: free TFs associate with rate constant $k_{on}$ to and dissociate with rate constant $k_{off}$ from an apparent number of chromatin binding sites $D_0$. The concentrations of all species involved depend on their quantities and the nuclear volume $V_n$. Inset: formula describing the chromatin-bound fraction of TFs that does not depend on the concentration of TFs

showed signs of binding site saturation (see below). For mEos2-TBP, we observed preferential binding to promoters of genes expressed in the early embryo[4] as compared to a genomic control by ChIP-qPCR (Fig. 2d and Methods). mEos2-Sox19b bound preferentially to Sox19b-specific target sites as compared to a genomic control (Fig. 2e and Methods). Injection of mEos2-TBP did not affect the development of embryos into prim-5 stage at 24 hpf (Fig. 2f and Supplementary Figure 3). For mEos2-Sox19b, no developmental phenotype was visible up to sphere stage, thus suggesting normal behavior during our measurement period up to oblong stage (Fig. 2f and Supplementary Figure 3). Thereafter, mEos2-sox19b injected embryos displayed developmental delay but no overt phenotype at mid-somitogenesis stages. At the prim-5 stage defects in development of posterior structures became obvious. The developmental delay observed for embryos into which Sox19b mRNA was injected is comparable to previous observations with embryos into which Sox3, which is functionally related to Sox19b, was injected[38]. Overall, these experiments indicate that chromatin-binding of mEos2-TFs approximate binding of endogenous TF within the limitations imposed by ectopic expression and an added fluorescent tag.

We could clearly detect the fluorescent signal from individual mEos2-TF molecules after photoswitching and exciting mEos2 with 405 nm and 561 nm lasers in a thin plane of a cell nucleus (Fig. 2g, Supplementary Figure 4 and 5 and Supplementary Movies 1–4). eGFP-Lap2β was coinjected as marker of the nuclear envelope[39] (Supplementary Figure 6 and see Methods). A drop of fluorescent signal to the background level in a single step is a common criterion indicating the single molecule nature of detected fluorescent protein signals (Fig. 2h)[40]. The signal-to-noise ratio (SNR) of individual detected mEos2 molecules was ca. 5 at the surface of the animal cap (Fig. 2i), comparable to signals of single fluorescent proteins in live mammalian cells[21]. SNR decreased to 2–3 at a distance up to approx. 60 μm in the embryo, thus approaching the limit of <2 of reasonable single molecule detection.

**TFs dissociate from DNA with two dissociation rate constants**. Having established single molecule imaging in living zebrafish embryos, we strived to quantify different parameters entering a description of TF-chromatin binding based on the law of mass action (Fig. 1b).

We first characterized the dissociation rate constants of mEos2-TBP and mEos2-Sox19b from chromatin using time-

lapse imaging (Fig. 3a and see Methods). This illumination scheme is able to resolve different dissociation rate constants and to separate them from the photobleaching rate constant by introducing dark times of different duration between two images, which reduces photobleaching but does not alter dissociation (Fig. 3b)[21]. We imaged bound mEos2-TF molecules in oblong embryos, where long cell cycles facilitated applying several time-lapse conditions compared to earlier stages. For each time-lapse condition, we monitored the duration a bound fluorescent molecule was visible (fluorescent "on" time) and collected these times in binding time histograms (Fig. 3c, d). As criterion for chromatin binding we required a mEos2-TF fusion protein to be localized within 0.2 μm² for at least 100 ms (see Methods)[20,21]. The histograms of both mEos2-TBP and mEos2-Sox19b followed a bi-exponential decay rather than a mono-exponential decay, revealing two distinct dissociation rate constants, respective residence times, for each factor (Fig. 3e and f and Methods).

A majority of mEos2-TBP molecules interacted transiently with chromatin with an average residence time of $(0.3 \pm 0.1)$ s (fit and corresponding error), while the remaining fraction of molecules bound stable to chromatin with a long residence time of $(6.8 \pm 0.7)$ s (fit and corresponding error). The photo-bleaching rate constant was $(7.3 \pm 0.2)$ s$^{-1}$ (fit and corresponding error). Our results are consistent with previous estimates of the TBP residence time on the order of seconds in living yeast[41], but do not show a very long residence time on the order of minutes calculated in another report[42] or seen in mammalian cells[35,37]. The discrepancy likely arises due to differences in TBP binding between species, or due to fast cell cycles in early zebrafish development. Fast TBP cycling is compatible with measurements of RNA Polymerase II dynamics in live cells that suggest fast transcription initiation times of ~8s[43]. Our measurements do not provide a hint for minute-long binding of molecules in interphase cells[44]. However, we cannot completely exclude a potential contribution of drift and out-of-focus movement of bound molecules during dark times to the observed residence time, which might lead to a potential underestimation of residence times.

For mEos2-Sox19b, we found a transient residence time of $(0.3 \pm 0.1)$ s (fit and corresponding error) for most molecules and a longer residence time of $(2.0 \pm 0.2)$ s (fit and corresponding error) for the remaining fraction. The photo-bleaching rate constant was $(5.8 \pm 0.2)$ s$^{-1}$ (fit and corresponding error). Compared to residence times of 12–15 s measured for Sox2 in mammalian cells[45], Sox19b

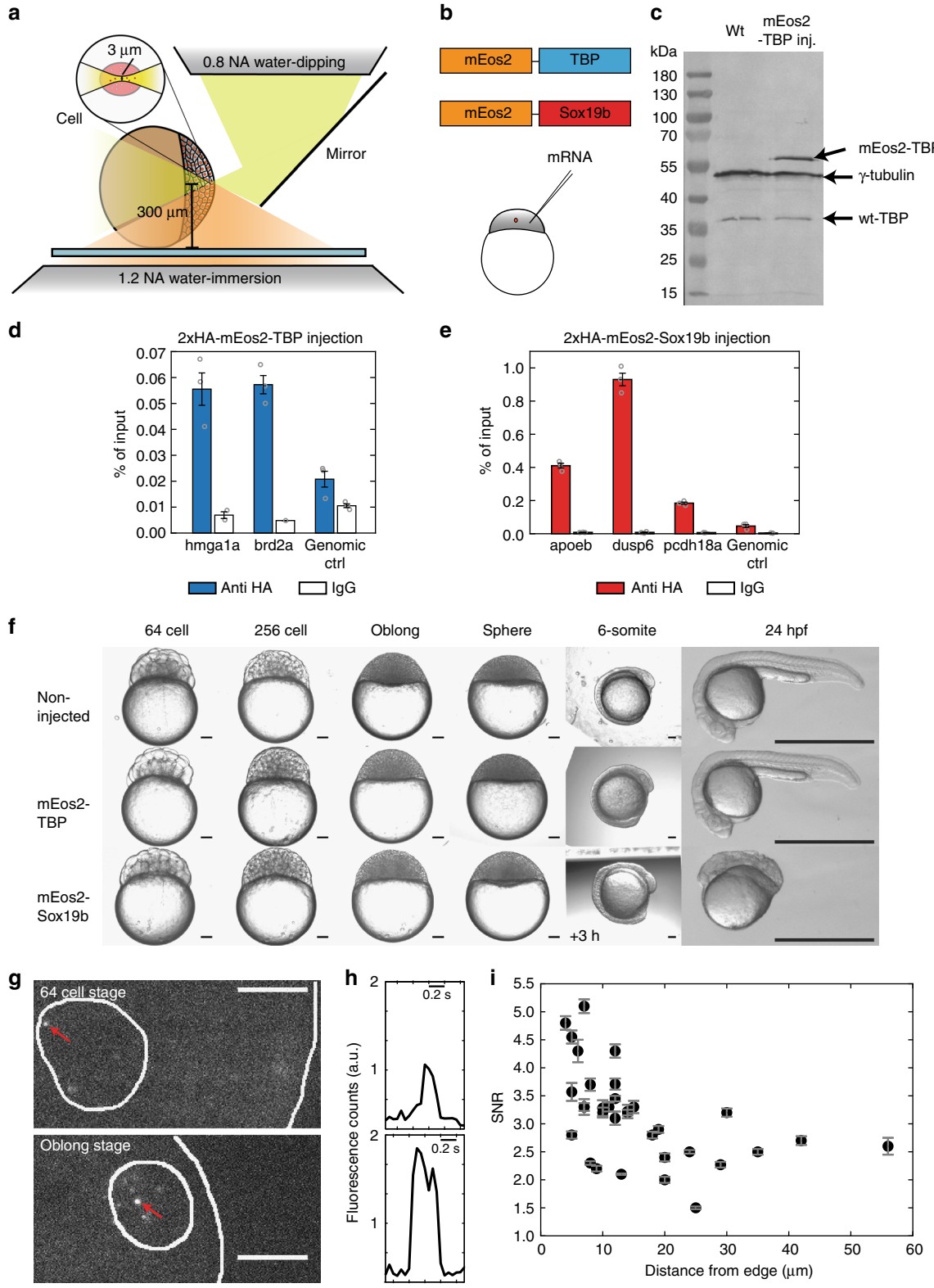

in zebrafish embryos binds shorter. This probably reflects differences in chromatin binding between Sox19b and Sox2, or might reflect differences between animal species.

A biphasic kinetic behavior similar to our observations for mEos2-TBP and mEos2-Sox19b has previously been observed for various other TFs and was identified with transient binding to unspecific sequences and binding to specific target sequences on

chromatin[45–49]. These previous observations suggest that transient interactions of mEos2-TF can be assigned to unspecific binding and long, stable interactions to specific binding on chromatin. For the quantitative analysis of mEos2-TF binding to chromatin we included transient and stable chromatin residence times in the TF-chromatin interaction model (Supplementary Figure 7).

**Fig. 2** Single molecule imaging in live zebrafish embryos by RLSM. **a** Sketch of reflected light-sheet microscopy of a zebrafish embryo. Inset: close-up view of a single cell within the embryo. **b** Sketch of mEos2-TBP and mEos2-Sox19b fusion proteins. mRNA coding for the constructs was injected into the 1-cell stage embryo. **c** Western blot analysis of TBP in wild type (wt) embryos (left lane) and injected with mEos2-TBP (right lane). γ-tubulin was used as loading control. Expected sizes are 50 kDa (γ-tubulin), 33 kDa (wt-TBP) and 59.4 kDa (mEos2-TBP). **d** ChIP-qPCR analysis of 2xHA-mEos2-TBP (blue, 3 technical replicates) binding to chromatin in the promoter region of genes *hmga1a*, *brd2a* and a genomic control and corresponding IgG controls. Values are given in % of input (mean ± s.e.m). **e** ChIP-qPCR analysis of 2xHA-mEos2-Sox19b (red, 3 technical replicates) binding to chromatin in the promoter region of genes *dusp6*, *apoeb*, *pcdh18a* and a genomic control and corresponding IgG controls. Values are given in % of input (mean ± s.e.m.). **f** Effect of mEos2-TBP and mEos2-Sox19b expression in embryos grown at 22 °C (up to 6-somite stage) or at 25 °C until 24 hpf. The scale bar is 100 μm up to the 6-somite stage and 1 mm at 24 hpf. **g** Fluorescence images of a 64-cell stage embryo and an oblong stage embryo expressing mEos2-TBP. The surface of the animal cap and the outline of the nucleus are indicated (white lines). Red arrows point to single mEos2-TBP molecules. The scale bar is 10 μm. **h** Time traces of mEos2-fluorescence of the molecules indicated in **g**. **i** Signal-to-noise ratio of single nuclear mEos2-TBP molecules as a function of nucleus distance from the surface of the animal cap. Values are calculated from >200 detected molecules within each nucleus (data from 3 embryos, mean ± s.e.m.)

**The chromatin-bound fraction of TFs increases during ZGA.** Next, we sought to quantify the bound fraction of mEos2-TFs and distinguish between transient and stable binding interactions at different stages during early development. The full assessment of dissociation rate constants requires measurements at many different time-lapse conditions[44], which is challenging during the fast cell cycles of the early embryo. Thus, we developed a suitable illumination scheme, interlaced time-lapse microscopy (ITM). In this illumination scheme, two subsequent image acquisitions are followed by a long dark time (Fig. 4a). By sorting detected molecules into different binding time classes, accounting for the exponential distribution of binding times and correcting for photobleaching (Fig. 4b), two interlaced time-lapse conditions are sufficient to obtain quantitative information on the molecule concentrations, the chromatin-bound fraction and the proportion of stable bound molecules (Supplementary Figure 8, Supplementary Figure 9, Supplementary Figure 10 and Supplementary Methods).

We used ITM to image mEos2-TBP and mEos2-Sox19b molecules between the 64-cell and the oblong stage and sorted molecules according to their binding time class (Fig. 4c, d). We then first tested whether our initial assumption that TFs bind to chromatin following the law of mass action was indeed true for mEos2-TBP and mEos2-Sox19b or whether effects of over-expression, which depend on the factor of interest[44,50–52], or facilitated dissociation would be visible. For each stage, we therefor plotted all chromatin-bound mEos2-TF molecules in an embryo versus all molecules detected in this embryo, utilizing small variations in mEos2-TF expression levels between embryos (Fig. 4c, d and Supplementary Figure 11). Within all stages, the relation between chromatin-bound and all mEos2-TF molecules was linear, as expected from the law of mass action. This indicates that saturation effects and facilitated dissociation are not observable, since both effects would yield a concave curve. The slope of the relation between chromatin-bound and all molecules differed between different stages, indicating a change of the chromatin-bound fraction of mEos2-TF molecules during development.

Next, we quantified the chromatin-bound fraction of mEos2-TFs by dividing the number of molecules identified as bound by the number of all detected molecules. Since the number of detected mEos2-TF molecules does not report on endogenous TF concentrations, we did not directly evaluate measured protein concentrations (Supplementary Figure 12). In contrast, the chromatin-bound fraction of mEos2-TF molecules is independent of concentration if saturation does not occur and therefore may reflect the behavior of endogenous TFs, independent of differences in expression. Our quantification including the linear corrections due to exponential binding time distributions and photobleaching revealed that the chromatin-bound fraction of mEos2-TBP increased ~6-fold and that of mEos2-Sox19b

increased ~2-fold between the 64-cell and the oblong stage (Fig. 5a, Supplementary Figure 13 and Supplementary Methods). This increase is in accordance with the expectations arising from a decreasing nuclear volume and a potential increase in chromatin binding sites (Fig. 1b).

Using the data recorded by ITM, we also quantified the stable bound proportion of chromatin-bound molecules for mEos2-TBP and mEos2-Sox19b by dividing the number of stable bound molecules by the number of all chromatin-bound molecules and correcting for exponential binding time distributions and photobleaching (Fig. 5b, Supplementary Figure 13 and Supplementary Methods). We found that this value was constant to good approximation for both TF species. ITM is sensitive to a change in the relative number of transient and stable binding events on chromatin, and to a change in the duration of stable binding interactions (Supplementary Methods). Thus, a constant proportion of stable bound molecules indicates that the dissociation rate constants of both mEos2-TF species do not change considerably during early embryo development.

**TF-chromatin associations correlate with nuclear volume.** Besides being dependent on dissociation rate constants, binding of a TF to chromatin is also dependent on the volume of the nucleus, the association rate constant and the number of accessible binding sites (Fig. 1b). To obtain a quantitative understanding of the increasing chromatin-bound fraction of mEos2-TFs, we thus next quantified the nuclear volume between the 64-cell and the oblong stage, based on the signal of eGFP-Lap2β (Fig. 5c and Supplementary Notes, Section 4.4.2). We confirmed that nuclear size did not change in mRNA-injected embryos compared to wild type embryos (Supplementary Figure 14). We found that nuclear volume decreased during this developmental period, as reported previously[1].

There is currently no experimental assay allowing measuring the association rate constant and the number of accessible DNA binding sites in a living embryo. We thus combined both parameters in the apparent number of chromatin binding sites and extracted this quantity as free parameter from the model of TF-chromatin interactions by inserting our measured chromatin-bound fraction of mEos2-TF molecules, dissociation rate constants and the nuclear volume (Fig. 5d and Supplementary Notes, section 4.4). For mEos2-TBP, we found that the apparent number of chromatin binding sites increased ~3-fold between the 64-cell and the oblong stage. In contrast to mEos2-TBP, for mEos2-Sox19b the apparent number of chromatin binding sites stayed approximately constant between the 64-cell and the oblong stage. This indicates that the decreasing nuclear volume dominates the increase in chromatin-bound fraction of mEos2-Sox19b.

The quantitative analysis of our data suggests a potential mechanism for changes in TBP and Sox19b associations with

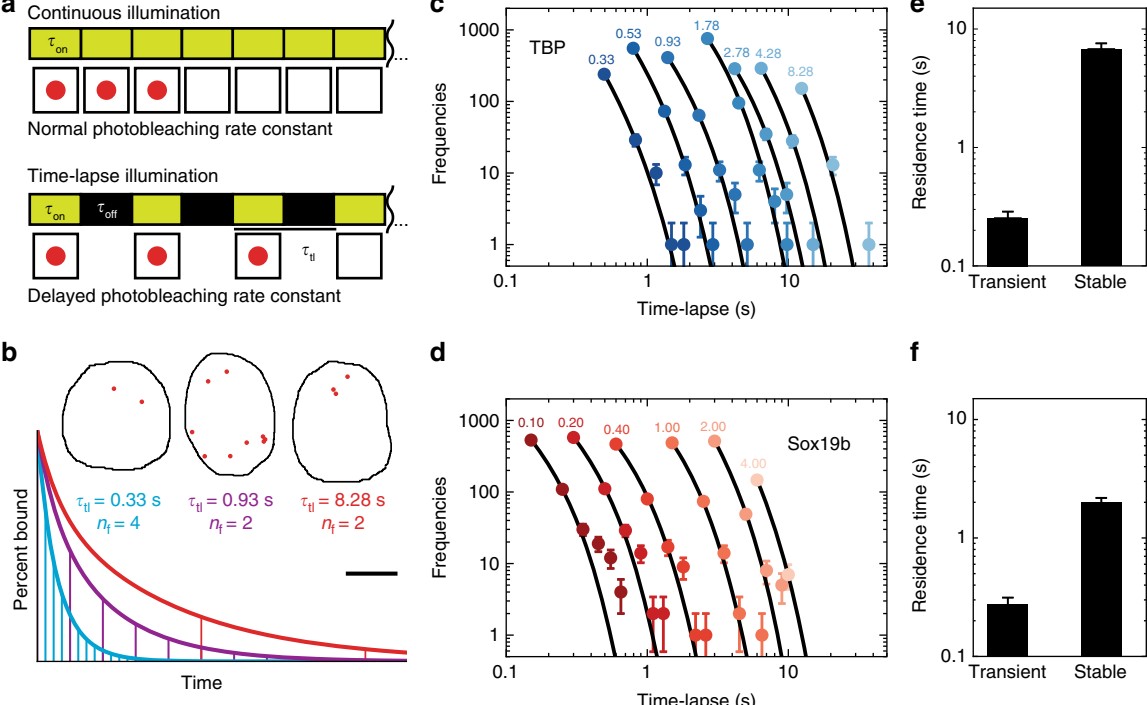

**Fig. 3** mEos2-TFs exhibit two dissociation rate constants. **a** Scheme of time-lapse microscopy. Red spheres indicate a detected mEos2-TF molecule. $\tau_{on}$ : time laser is on, $\tau_{off}$ : time laser is off, $\tau_{tl}$ : time-lapse time. **b** Schematic decay of visible fluorescence over time under short (blue), intermediate (magenta) and long (red) time-lapse intervals indicated by the horizontal lines. Inset: Measured mEos2-TBP molecules still bound after 4 short (left), 2 intermediate (middle) and 2 long (right) time-lapse intervals demonstrating effects of photobleaching (left) and dissociation (right). Starting number were 100 molecules. The scale bar is 5 μm. **c, d** Histograms of fluorescent "on" times at different time-lapse conditions of **c** mEos2-TBP and **d** mEos2-Sox19b in nuclei of oblong embryos (mean ± s.d.). Time-lapse intervals are written on top of the data points. Lines: global fit of a bi-exponential decay model (Equation 1 in Methods). Data includes 4780 molecules from 15 embryos for mEos2-TBP and 3330 molecules from 4 embryos for mEos2-Sox19b. For clarity, only the first six points of histograms are shown. **e, f** Residence times of **e** mEos2-TBP and **f** mEos2-Sox19b. Errors are the s.d. from the fits in **c** and **d**

chromatin during the developmental period embracing ZGA (Fig. 5e), inspired by previous reflections on the importance of nuclear size and concentration:[53–55] both TBP and Sox19b search for their specific target sequences, associated with long residence times, while scanning chromatin with transient interactions. For TBP, a combination of decreasing nuclear size, which influences DNA concentration, and an increase in the apparent number of chromatin binding sites ensures that increasingly higher fractions of TBP are associated with chromatin during this important developmental period in the early embryo. In contrast, the increase in chromatin-bound fraction of Sox19b molecules seems to be mainly enhanced as result of the decreasing nuclear size.

## Discussion

In principle our modifications to the RLS microscope allow movement of the light-sheet focal plane to any position within the embryo. In practice, single molecule imaging is limited in height above the sample surface by the working distance of the detection objective, which in our case was ~300 μm. Single molecule imaging in the embryo was limited to a depth of approx. 60 μm from the surface of the animal cap. This limit is due to absorption and scattering of fluorescent light within the tissue of the embryo[56]. Once the cell radius decreased below this threshold, single molecule imaging in live growing zebrafish embryos became straightforward using RLS microscopy. Given the relatively large size of zebrafish embryos compared to other model systems, RLS microscopy might very well be suited to image single fluorescent molecules also in other model systems such as *C. elegans*, *D. melanogaster*[27], in vitro grown mouse blastocysts or organoids.

Interlaced time-lapse microscopy (ITM) is an illumination scheme, which we developed to quantify TF-chromatin interactions when overall acquisition time is limited, for example in the developing embryo. It relies on time-lapse imaging of the sample with intermingled short and long dark times. ITM yields quantitative information on concentrations of fluorescently tagged molecules, the chromatin-bound fraction of a TF and the relative numbers of transient and stable chromatin-bound TF molecules. Using ITM, these quantities can conveniently be compared between different conditions such as developmental stages. While the raw parameters contain already important relative information, they become even more meaningful if dissociation and photobleaching rate constants are known in one condition. In this case a linear correction accounting for an exponential decay of interaction counts with time and for photobleaching can be applied. Importantly, ITM enables a quantitative assessment on changes in transient and stable bound fractions or residence times between different conditions. ITM is currently limited to two distinct binding time classes. Furthermore, binding time classes can only be resolved if they differ by roughly one order of magnitude in binding time.

We performed single molecule tracking experiments to monitor how binding of mEos2-TBP and mEos2-Sox19b to chromatin evolved during early embryo development. For both TF species, we observed that the percentage of chromatin-bound molecules increased. The relation between chromatin-bound and all detected TF molecules was linear in every developmental stage, indicating that binding to chromatin is well described by the law of mass action within each stage.

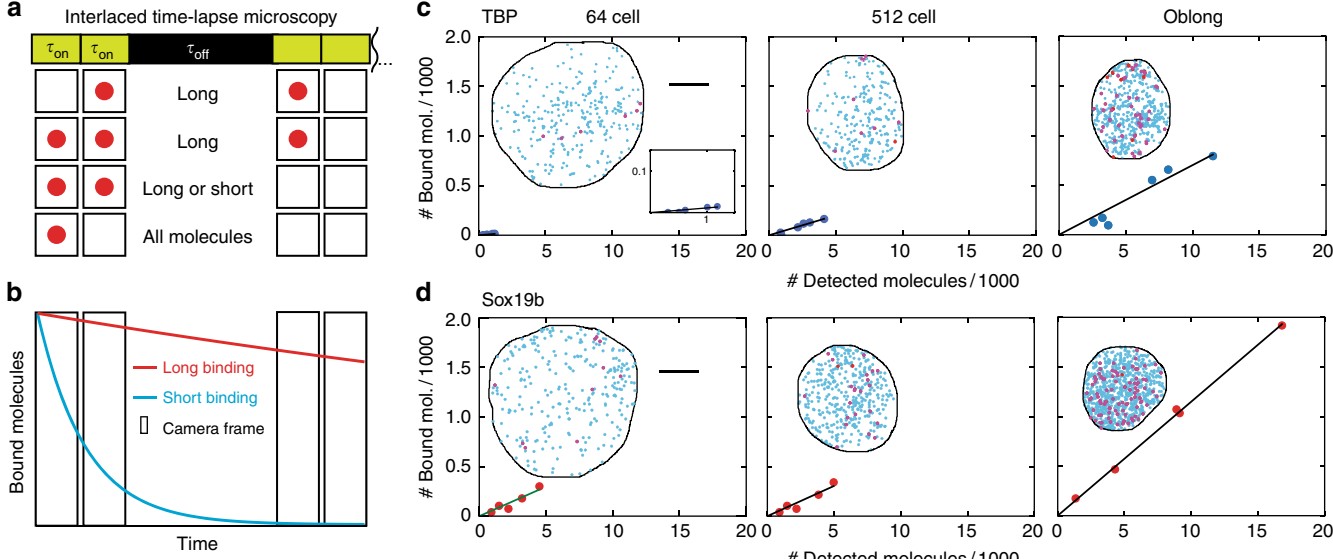

**Fig. 4** mEos2-TFs bind according to the law of mass action. **a** Scheme of interlaced time-lapse microscopy. Red spheres indicate a detected mEos2-TF molecule. $\tau_{on}$: time laser is on, $\tau_{off}$: time laser is off. **b** Schematic decay of visible fluorescence over time for the short (blue) and the long (red) bound fraction of molecules. **c**, **d** Number of molecules classified as bound compared to all detected molecules at the indicated stages. Every dot represents one embryo injected with **c** mEos2-TBP and **d** mEos2-Sox19b. The panel includes in total 53504 molecules from 5 embryos (64-cell stage) and 6 embryos (later stages) for mEos2-TBP and 76407 molecules from 5 embryos for mEos2-Sox19b. Insets: time-projection of all molecules within a nucleus classified as unbound (blue), short bound (magenta) and long bound (red). The scale bar is 5 μm

When inserting the measured values for nuclear size and dissociation rate constants in the model for TF-chromatin interactions, we found that for mEos2-TBP, in addition to the decreasing nuclear size, an increase in the apparent number of chromatin binding sites accounted for the increasing chromatin-bound fraction. This is in accordance to observations of increasing DNA accessibility in the development of other species[6,7]. However, caution has to be taken when comparing the fold-increase in apparent accessible binding sites of our in vivo single molecule tracking measurement with measurements based on DNase I hypersensitivity[7] or transposase accessibility[6]. While the latter techniques reveal regions of nucleosome-free DNA, the apparent number of chromatin binding sites we determined combines different parameters affecting TF-chromatin associations. Besides a potential change in accessible DNA binding sites, it might also include effects such as changes in the association rate constant and a decrease in binding competition between TFs and histones[3,9] or other chromatin-binding molecules. More studies will be necessary in the future to further disentangle these contributions.

Interestingly, for mEos2-Sox19b, the apparent number of chromatin binding sites did not change during early development. Thus, in contrast to TBP, our data indicate that for Sox19b the chromatin-bound fraction is mainly enhanced due to a decrease in nuclear volume. It is tempting to speculate that a reason to this difference might be the potentially different modes of binding of TBP and Sox19b to DNA. TBP binds to nucleosome-free DNA[29]. Sox19b shares a similar DNA binding domain with Sox2[57], is functionally related to Sox2[32] and plays a major role in early zebrafish development[33,34]. Thus, similar to Sox2 in mammalian cells, it might act as pioneering factor[58,59] in early zebrafish development and be able to recognize and bind to its target site also if it is occupied by nucleosomes[60]. In this case, the apparent number of chromatin binding sites for Sox19b would not need to increase even if the amount of nucleosome-free DNA increased. Our observations raise the intriguing yet speculative possibility that the shrinking nuclear volume, by

enhancing the percentage of chromatin-associated Sox19b and other potential pioneering factors, may contribute to setting the initial stage for ZGA in zebrafish embryos. To which degree pioneering TFs[33,58,60] or chromatin remodelers[61], that have been shown to be essential in mouse embryogenesis[62], might modulate the accessibility of DNA to TFs such as TBP and thereby contribute to the association of the transcription machinery during early zebrafish development will be important to solve in the future.

In contrast to the chromatin-bound fraction in absence of saturation, the absolute number of chromatin-bound TF molecules depends on the concentration of TF in the nucleus. As numerical example, if initially 10 molecules were present in a nucleus at a bound fraction of 0.1, 1 molecule would be bound. If the level of the TF increased during development, for example to 100 molecules, already 10 molecules would be bound. As binding of a TF is an essential step in transcription initiation of most genes, it is the absolute number of bound molecules that probably is correlated to transcriptional activity. Questions related to the regulation of transcriptional activity in the embryo thus need to address the number of maternally inherited TF molecules and mRNAs initially present in the zebrafish embryo[63] as well as the kinetics of TF translation[30] in addition to nuclear size or DNA accessibility. In this view, an increase in the chromatin-bound fraction of TFs contributes to achieving a high number of chromatin-bound TFs at later developmental stages by allowing the cell to more efficiently allocate TFs to the bound state. In our example above, if during development also the bound fraction increased, say to 0.4, 40 molecules instead of just 10 would be bound.

Our experiments emphasize the role of the nuclear size for a quantitative description of TF-chromatin interactions. Since changes in nuclear size globally affect the concentration of DNA in the nucleus, it seems likely that other DNA-binding factors are modulated in their binding behavior to DNA similar to TBP and Sox19b. We note that for each TF specific biological constraints, such as formation of local high density hubs[64] need to be

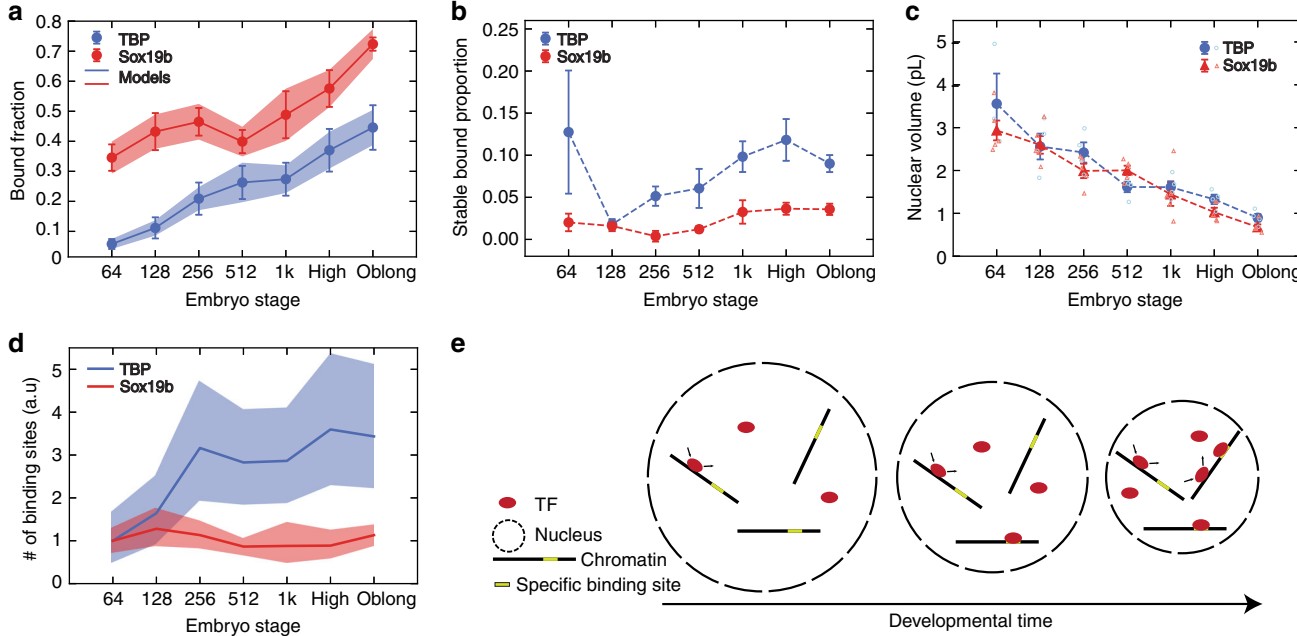

**Fig. 5** The chromatin-bound fractions of mEos2-TFs increase during early development. **a** Chromatin-bound fraction of mEos2-TBP (blue) and mEos2-Sox19b (red) during development (calculated from data presented in Fig. 4 and Supplementary Figure 11, mean ± s.e.m). Lines represent the chromatin-bound fraction as calculated from the law of mass action using data from **b** and **c** and extracting the apparent number of chromatin binding sites as free parameter. Shades represent error intervals propagated from the errors in stable bound proportion and nuclear volume. The panel includes in total 120572 molecules from 5 embryos (64-cell stage) and 6 embryos (later stages) for mEos2-TBP and 158912 molecules from 5 embryos for mEos2-Sox19b. **b** Stable bound proportion of chromatin-bound mEos2-TBP (blue) and mEos2-Sox19b (red) molecules during development (calculated from data presented in Fig. 4 and Supplementary Figure 11, mean ± s.e.m.). The panel includes in total 5677 molecules from 3 embryos (64-cell stage), 4 embryos (128-cell stage), 5 embryos (256-cell stage) and 6 embryos (later stages) for mEos2-TBP and 13103 molecules from 5 embryos for mEos2-Sox19b. The dashed line is given as guide to the eye. **c** Nuclear volume of mEos2-TBP (blue spheres) and mEos2-Sox19b (red triangles) injected embryos during development. Data includes 5, 16, 22, 25, 32, 37, 84 measurements from 3 embryos (64-cell stage) and 4 embryos (later stages) for mEos2-TBP and 17, 21, 33, 60, 42, 90, 98 measurements from 5 embryos for mEos2-Sox19b (mean ± s.e.m. mean values for single embryos shown as open circles for TBP and open triangles for Sox19b). The dashed line is given as guide to the eye. **d** Apparent number of chromatin binding sites of mEos2-TBP (blue) and mEos2-Sox19b (red) as calculated from the law of mass action using data from **a**, **b**, and **c**. Shades represent error intervals propagated from the errors in chromatin-bound fraction, stable bound proportion and nuclear volume. **e** Scheme of the concentration model. During early embryo development the size of individual nuclei decreases, thereby increasing the concentration of DNA and increasing the chromatin-bound fraction of TFs

considered in addition to the effect of nuclear size. This is particularly important in the context of ZGA, since the decrease of nuclear size coincides with the temporal onset of ZGA. While the basis of nuclear size regulation is not yet well understood, scaling between cell size and nuclear size appear to be a widespread phenomenon and several principle mechanisms and possible regulators have been suggested[65,66]. Nuclear size may be regulated by a limiting cytoplasmic pool of building subunits[53] such as components of the nuclear envelope[67] or the nuclear pore complex[68], by nuclear import[69] or a force balance between cytoskeletal and nuclear components[70].

In *Xenopus*, the effect of nuclear size and the nucleo-cytoplasmic volume ratio on timing of ZGA has been investigated[71]. Jevtic et al. artificially increased or decreased nuclear size by adding reticulon or other factors and observed premature activation or delay of zygotic transcription when nuclear size was increased or decreased[71]. These experiments do not contradict our measurements that revealed an increased chromatin-bound fraction of TFs when nuclei were small. Increasing the nucleo-cytoplasmic volume ratio of individual cells as done by Jevtic et al. might change the concentrations of a TF or a repressor of TF binding[8,72], or otherwise change chromatin structure compared to an unchanged reference cell of the same stage. In effect, the apparent number of chromatin binding sites and the absolute number of bound TFs might be increased, leading to premature transcription.

While differing in the details, other species including *Drosophila*, *Xenopus*, sea urchins and *C.elegans* share common characteristics with zebrafish during early stages of development. Their embryonic genomes are transcriptionally silent for several cell division cycles[73]. In these species, the size of individual nuclei decreases considerably during the initial phase of rapid cell divisions before ZGA[8,74–76]. Such a decrease is also observed in mammals[77]. It is thus very likely that a similar concentration mechanism increasing the chromatin-bound fraction of TFs and thus potentially facilitating transcription onset in zebrafish embryos also applies to other species.

## Methods

**Reflected light-sheet (RLS) microscopy.** To image live zebrafish embryos, we modified the reflected light-sheet microscope originally designed to image live cells[21].

Laser light of 405 nm (Laser MLD, 200mW, Cobolt, Solna, Sweden), 488 nm (IBEAM-SMART-488-S-HP, 200 mW, Toptica, Graefelfing, Munich, Germany) and 561 nm (Jive 300 mW, Cobolt) was expanded to 1.2 mm beam diameter by Kepler telescopes with lenses of a focal length of 75 mm and 100 mm for the 488 nm laser and of 60 mm and 150 mm for the other lasers. Afterwards it was combined using dichroic mirrors (F48–486, F43–404, AHF, Tübingen, Germany) and selectively filtered by an AOTF (AOTFnC-400.650-TN, AA Optoelectronics, Orsay, France). Subsequently, light was coupled into a single-mode fiber (S405, Thorlabs, Dachau, Germany) by a fiber-coupler (60FC-4-RGBV11–47, Schäfter +Kirchhoff, Hamburg, Germany). After the fiber, the beam was expanded by a fiber collimator (60FC-T-4-RGBV42–47, Schäfter+Kirchhoff) to a diameter of 1.5 cm and unilaterally focused by a cylindrical lens ($f = 15$ cm, ACY254–150-A, Thorlabs) into the back-focal plane of a water dipping objective (40× 0.8 NA HCX

Apo L W, Leica, Wetzlar, Germany). The effective NA was reduced by an iris after the cylindrical lens and setting a beam diameter of 1.5 mm before the water dipping objective. The light-sheet formed by the objective was reflected by the chip of an AFM cantilever (HYDRA2R-100N-TL-20, AppNano, Mountain View, CA) that was coated with 4 nm Titanium and subsequently with 40 nm Aluminum by physical vapor deposition (PVD). The resulting light sheet had a thickness of approximately 3 μm. Bright field illumination was achieved by a LED (720 nm, HCA1 H720, 70 mW, Roithner, Vienna, Austria) whose light was overlaid with the laser light before the illumination objective by a dichroic mirror (F38–635, AHF).

The cantilever used as micro-mirror was mounted on a xyz-micro-positioning stage (Witech, Ulm, Germany) to enable precise remote-controlled positioning of the cantilever. The whole composition of bright-field illumination, fiber-collimation, cylindrical lens and illumination objective was mounted onto a rotatable stage (XYR1/M, Thorlabs) positioned on a tripod which was mounted on a commercial microscope body (TiE, Nikon, Duesseldorf, Germany). The sample dish was mounted on a piezo z-stage with 500 μm travel (Nano-ZL 500, Mad City Labs, Madison, WI).

Fluorescence light was collected with a water-immersion objective (60x 1.20 NA Plan Apo VC W, NIKON) filtered by a dichroic mirror (F73–866/F58–533, AHF), an emission filter (F72–866/F57–532, AHF) and a notch filter (F40–072/F40–513, AHF). before being post-magnified (1.5×) and detected by an EM-CCD Camera (iXon Ultra DU 897U, Andor, Belfast, UK).

All electronic parts of the setup were controlled using NIS Elements software (Nikon) and a NIDAQ data acquisition card (National Instruments, Austin, TX). Macros were written in a C-like NIS Elements macro language.

**Fluorescent fusion proteins.** A pCS2+ plasmid backbone containing mEos2-TBP (zebrafish TBP, uniprot ID Q7SXL3 and mEos2, uniprot ID Q5S6Z9, both separated by a SGSGSG linker) was a gift from Nadine Vastenhouw (Max Planck Institute of Molecular Cell Biology and Genetics, Dresden, Germany). A pCS2+ plasmid backbone containing eGFP-Lap2β was a gift from Nadine Vastenhouw and originates from Marija Matejcic from the Caren Norden lab (Max Planck Institute of Molecular Cell Biology and Genetics, Dresden, Germany).

The Sox19b (uniprot ID Q9DDD7) sequence was extracted from 1-cell stage zebrafish cDNA as previously described[78] using the SuperScript VILO cDNA synthesis kit (Invitrogen). PCR was done with Q5 High-Fidelity DNA Polymerase (New England BioLabs) with addition of GC enhancer using the primers listed in Supplementary Table 1. To generate a mEos2-Sox19b fusion protein, we exchanged TBP in pCS2+ with Sox19b using restriction enzymes BspEI and XbaI (New England BioLabs). For 2xHA-mEos2-TBP and 2xHA-mEos2-Sox19b, mEos2 in pCS2+ was exchanged for 2xHA-mEos2 using restriction enzymes BamHI and BspEI (New England BioLabs).

**mRNA synthesis.** For mRNA synthesis, 3 μg of the plasmids were linearized with NotI-HF (New England BioLabs). mRNA synthesis was started according to mMESSAGE mMACHINE SP6 Transcription Kit (Ambion). Incubation time was elongated to 4 h. Afterwards the newly synthesized mRNA was cleaned up according to RNeasy Plus Mini kit (Qiagen).

**Western blot.** We loaded 7.6 or 15.2 μg of total protein isolated from 100 to 200 1k-cell stage zebrafish embryos on a 10% SDS gel. We used a standard Western Blot protocol with the antibodies listed in Supplementary Table 3. The TBP-antibody was diluted between 1:500 and 1:250, the gamma-tubulin antibody was diluted between 1:100,000 and 1:10,000. Secondary antibody was diluted 1:10,000. To visualize binding of the secondary antibody, we incubated the blot with BCIP-NBT (ChemCruz). For quantification of overexpression we used the ImageJ 1.52e gel band analysis plug-in. We divided the integrated intensity of the above-background signal of ectopic and endogenous TBP bands to estimate overexpression levels.

**Staging of embryos.** We imaged embryos using an inverted bright field microscope or upright stereomicroscope from Leica. We followed the development of embryos injected with mRNA coding for mEos2-TF from the 16-cell stage onward at 25 °C. Since single molecule measurements were performed at 22 °C, we confirmed comparable development at this temperature and report representative images up to 6-somite stage at 22 °C.

**ChIP-qPCR.** To design primers for TBP Chip-qPCR, DNA sequences of hmga1a and brd2a were extracted from UCSC Genome Browser (zebrafish Sep. 2014, GRCz10/danRer10). Primers were designed to bind within a 200 bp window upstream the genes and to yield PCR products of about 180bp in length. Primers for Sox19b Chip-qPCR for apoeb, dusp6, and pcdh18a were taken from publications as indicated in Supplementary Table 2. As genomic negative control, we designed primers for tbx16 (in exon 9 of tbx16, GRCz10/danRer10). All primers had a reasonable E-value with no other or few additional hits in BLASTN (Danio rerio, Ensembl). All primers are listed in Supplementary Table 2.

ChIP was performed as described in ref. [49] with modifications for zebrafish embryos. For 2xHA-mEos2-TBP around 300 embryos (early 1k-cell stage) per IP and for 2xHA-mEos2-Sox19b around 250 embryos (early 1k-cell stage) were fixed

in 1.85% formaldehyde at RT for 15 min After nuclear lysis, DNA was sheared to fragments between 200 and 600 bp using a Diagenode Bioruptor UCS-200. For IP of 2xHA-mEos2-TBP and 2xHA-mEos2-Sox19b we used 3 μg of anti-HA (Abcam: ab9110) and for mock IP 3 μg of normal rabbit IgG (Cell Signaling: #2729). After recovery and purification, DNA was analyzed by qPCR with Power SYBR Green PCR Master Mix (Applied Biosystems) in a StepOne Plus Real-Time PCR System (Applied Biosystems). All ChIP results were normalized to the amount of input DNA. Experiments were performed in three technical replicates. Values are given in % of input (mean ± s.e.m). Undetermined values in the IgG control were not set to zero but omitted to avoid an underestimation of background.

**Sample preparation.** We dechorionated wild-type AB zebrafish embryos directly after fertilization and synchronized them by visual selection. For mRNA injection, we set the droplet diameter to (63 ± 3) μm by adjusting the injection duration. This corresponded to a droplet volume of 1 nl varying by up to ±15%. We injected 60pg mEos2-TBP mRNA or 29 pg mEos2-Sox19b mRNA respectively together with 9pg eGFP-Lap2β mRNA in 1 nl into the animal cap at the 1-cell stage to visualize the transcription factor and the outline of the nucleus[39]. Embryos developed to the 32-cell stage at 28 °C. At the 32-cell stage, we mounted embryos on a glass bottom dish (Delta T, Bioptechs, Butler, PA) at room temperature. During fluorescence imaging, we counted cell cycles based on the breakup and re-formation of the nuclear envelope labeled with eGFP-Lap2β. The embryos developed to sphere stage within 4.5 h, after which fluorescence imaging was stopped.

**Data acquisition.** For time-lapse microscopy, we took videos of mEos2 fluorescence after an initial period of photo-activation of 0.5 s. We set illumination time $\tau_{on}$ with 561 nm laser light to 250 ms for mEos2-TBP and to 50 ms for mEos2-Sox19b. Dark times between illumination times were varied depending on the time-lapse condition. If nuclei drifted due to cellular or whole embryo movement, acquisition was interrupted.

For ITM, after an initial period of photo-activation of 0.5 s, two successive frames were taken during a total illumination time of 330 ms followed by a dark time of 750 ms. If nuclei drifted due to cellular or whole embryo movement, acquisition was interrupted.

**Data analysis.** We identified bright pixels as candidates for fluorescent molecules, if their gray value was above the mean plus 3.5 standard deviations of the whole image[21]. Based on simulated data of similar SNR and molecule densities including noise, we estimated a detection error of ca. 10%[79]. We fitted a 2D-Gaussian curve to the surrounding area of the pixels. The center determined by the fitting function was taken as position of the fluorescent emitter. We connected fluorescent spots to tracks according to distance criteria. A spot that was localized within 0.2 μm² for at least 100 ms in time-lapse microscopy was identified as a bound molecule[20,21]. The effective pixel size in the specimen was measured to be 166 nm. For these steps of analysis, programs written in MATLAB (MathWorks, Natick, MA)[21] were used. Subsequent analysis steps were written in Python 2.7.9.

To quantify nuclear concentrations of mEos2 fusion proteins we photo-activated the sample several times per cell cycle with a lag time of approx. 2 min to allow for sufficient recovery of unactivated molecules. We analysed only the first frame after photo-activation.

To extract molecular DNA-residence times we implemented a global fitting approach in Python[21]. The numbers of frames a molecule was detected at the same position (times of visible fluorescence) were binned to histograms and a double-exponential distribution (Eq. 1) was globally fitted to the error-weighted histograms using a Levenberg-Marquardt least squares algorithm.

$$
\begin{aligned}
f_T(t) = A \cdot \Big[ & B \cdot \left( k_b \cdot \left( \tfrac{\tau_{on}}{\tau_{tl}} \right) + k_1 \right) \\
& \cdot \exp\left( - \left[ k_b \cdot \left( \tfrac{\tau_{on}}{\tau_{tl}} \right) + k_1 \right] \cdot t \right) + (1 - B) \\
& \cdot \left( k_b \cdot \left( \tfrac{\tau_{on}}{\tau_{tl}} \right) + k_2 \right) \\
& \cdot \exp\left( - \left[ k_b \cdot \left( \tfrac{\tau_{on}}{\tau_{tl}} \right) + k_2 \right] \cdot t \right) \Big]
\end{aligned}
\tag{1}
$$

Parameters optimized were the bleaching rate $k_b$, the off-rates of the bound TF molecules $k_1$ and $k_2$, the fraction of long and short bound molecules $B$ and $1 - B$ and the overall amplitude $A$. The parameters $\tau_{on}$ and the time-lapse time $\tau_{tl}$ (including illumination time and dark time) were pre-set and not subject to optimization. The stopping criterion was set to a relative change of $10^{-7}$ in the sum of error-squares. Reduced $\chi^2$-values were compared between a double- and a single exponential model. For the global fit of mEos2-TBP histograms, reduced $\chi^2$-values of 0.010 and 0.015 were found for the double-exponential and the single-exponential model respectively. For the global fit of mEos2-Sox19b histograms, reduced $\chi^2$-values of 0.012 and 0.015 were found for the double-exponential and the single-exponential model respectively. These values prefer the double-exponential model over the single exponential one.

In interlaced time-lapse microscopy (ITM), successive localizations of mEos2-TF within 166 nm were classified by their visible fluorescence time: Localizations surviving at least one dark time period were classified as long-binding events while

events appearing in at least two successive frames were classified as all binding events (long or short). We counted events for each stage in each embryo. If in a certain stage of an embryo only one molecule was bound, the stage was excluded from this analysis. Errors and means of these numbers between sets of embryos were determined by a bootstrapping procedure where a set of 900 random subsets, each containing 80% of the measured data, was analysed. The ratio of long to all binding events $R$ is not exactly equivalent to the ratio of stable to all bound molecules $B$ as bleaching and unbinding would lead to counting of stable binding events to the class of transient binding events. Dividing the number of bound molecules by the number of molecules that are detected at least in one frame yields the ratio $F$. This ratio can also be underestimating the true proportion of bound to freely diffusing molecules $D$. Thus, a correction formula was derived in mathematical detail in Supplementary Methods, section 3.3. Examples for the correction procedure are given in Supplementary Methods, section 3.3.

**Code availability**. The code to analyze ITM and time-lapse data is available as Python script on the Dryad Data Repository (https://dx.doi.org/10.5061/dryad.g154580)[80]. Other code used in this study is available from the corresponding author upon reasonable request.

## Data availability

Data supporting the findings of this manuscript are available from the corresponding author upon reasonable request. A reporting summary for this Article is available as a Supplementary Information file. All raw single particle tracking data is available in Matlab file format on the Dryad Digital Repository. In particular, the source data underlying Figs. 3c-f, 4c-d, 5a-d, Supplementary Figure 11, Supplementary Figure 12 and Supplementary Figure 13 are available on Dryad Digital Repository (https://doi.org/10.5061/dryad.g154580)[80].

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

## Acknowledgements

We thank Astrid Bellan-Koch (Institute of Biophysics, Ulm University, Ulm, Germany) for help with cloning and Western Blots, Nadine L. Vastenhouw, Shai R. Joseph and Máté Pálfy (Max Planck Institute of Molecular Cell Biology and Genetics, Dresden, Germany) for the introduction to zebrafish techniques, the mEos2-TBP and eGFP-Lap2β mRNA and plasmids (eGFP-Lap2β originates from Marija Matejcic, Caren Norden lab, Max Planck Institute of Molecular Cell Biology and Genetics, Dresden, Germany), discussions and comments to an early version of the manuscript. The work was in part funded by the German Research Foundation (No. GE 2631/1–1 to J.C.M.G. and No. 316249678, SFB 1279 and INST 40/493-1, SFB 1149 to G.W.), the European Research Council (ERC) under the European Union's Horizon 2020 Research and Innovation Program (No. 637987 ChromArch to J.C.M.G.), the European Union (ERA-CVD Cardio-Pro, BMBF No. 01KL1704 to G.W.), the German Academic Scholarship Foundation (to M.R.) and the Carl Zeiss Foundation (To A.P.P.). The authors thank the Ulm University Center for Translational Imaging MoMAN for its support.

## Author contributions

J.C.M.G. and M.R. designed experiments. M.R. constructed the reflected light-sheet microscope. M.R. performed experiments. A.P. performed cloning and designed primers for qPCR. A.P.P. and M.R. performed ChIP-qPCR experiments. C.J. and A.P. staged embryos. M.R., C.J. and A.P.P. performed Western Blots. M.R. and J.C.M.G. analyzed data. J.C.M.G. and G.W. supervised the work. J.C.M.G. and M.R. wrote the manuscript with comments from all authors.

## Additional information

**Competing interests:** The authors declare no competing interests.

