## [Peer Review File · Nature Communications]

Reviewers' Comments:

Reviewer #1:

Remarks to the Author:

This ms is an exciting report on the application of a novel single molecule imaging approach to detect TBP proteins in nuclei of developing zebrafish embryos. As a developmental biologist, I found the primary imaging data impressive which represents a breakthrough in real time in vivo imaging of transcription factor activity in living vertebrate embryos. The manuscript builds on the technology and demonstrates convincing correlation between binding of a misexpressed, tagged TBP to chromatin and dynamics of nucleus size during the cleavage stages prior to global genome activation. The observed binding profile suggests gradual increase of chromatin accessibility prior to ZGA and will be useful entry into revisiting the mechanisms of zygotic genome activation. However, in its current form the ms is hindered by overinterpretation of the findings and goes too far in concluding on the causative relationship between nucleus size and triggers of genome activation. The measurements are elegant, but remain correlative and do not provide sufficient lines of causative evidence (which one would be able to generate by interference experiments) to substantiate the dramatic claims presented in the title, abstract and subheadings of the Results section.

Besides these issues, there is a concern with the structure of the ms: lots of the novel technology and important sets of evidence are presented in the supplementary material for no obvious reason. The ms would be better suited to repitch as a primarily technology paper with elegant demonstration of utility by showing the TBP binding profiles and their correlation with nuclear size. The overemphasis on the interpretation of the underlying biology begs for additional experiments, and does not give justice to an otherwise excellent technological advance.

Major points

1. Single molecule imaging and detection of chromatin binding by overexpressed TBP

The imaging data is generated by a novel RLSM imaging approach and processed by innovative imaging and image processing pipeline which led to impressive and convincing observations on the level of detection and quantification of the single molecule binding signals. What is questionable is whether the data truly reflects endogenous TBP functions. Firstly, there is no information given about the key reagent (mEos2-TBP) of their study (apart from a statement that it has been a gift from a colleague and otherwise unpublished). This lack of information raises several questions: what is the origin (species) of the TBP used, how was the fusion with Eos2 designed, how was the fusion validated (does the fusion TBP still bind DNA with similar efficiency to wild type, does it remain biochemically and biologically functional, does its overexpression by microinjected mRNA affect development and timing of MBT, does it affect the measured cellular and cell cycle parameters (size, cycle length etc.) and how does its distribution and concentration compare to endogenous TBP? The reader is left to assume without experimental evidence whether the detected fusion molecule is representative for the distribution and behaviour of TBP proteins in early embryos.

Secondly, TBP is known to be highly dynamic in frogs and fish (publications by Veenstra and Muller labs), it is deposited maternally alongside 2 other TBP family members with distinct dynamics, but partially overlapping functions and post-transcriptionally regulated to dramatically increase protein levels just before MBT (Veenstra, Woolfe). The concentration of exogenous TBP after mRNA injection is unlikely to follow this pattern.

Thirdly, there seems to be albeit, not huge but consistent discrepancy between their model and measured TBP binding between 64 and 128 cell stages (Fig 3a and Fig4a) is this difference due to a factor that is not modelled by injected mRNAs excluded from posttranscriptional regulation (?). Such regulatory information is likely carried in the non-coding sequences of TBP, are these sequences present on the expression construct? Since we are not given information on the reporter expression construct it is unclear how much does this construct mimic endogenous TBP and its translational regulation.

The supplemental AVI files, show cytoplasmic foci similar to those in the nucleus. Can these be

explained?

2. Why do the authors ignore the nucleo-cytoplasmic ratio as parameter in their study and why do they suggest nuclear volume (without cytoplasm/cell volume) as an independent factor defining TBP concentration dynamics? Since the NC ratio is a well-known and very well established regulator of zygotic genome activation, and likely regulating chromatin accessibility (by for example the dilution of the repressive factor such as histones, see works by the Gurdon and Vastenhouw labs) one would expect that besides nuclear volume, cellular volume measurements are also carried out together with TBP binding to test if the fit is better with NC ratio than by nuclear volume only. Cell volumes are heterogeneous and change non-linearly between cell cycles (Kane and Kimmel, *Development* 1993). Moreover, the NC ratio dynamic appears to change dramatically in early *Xenopus* development (Jevtic and Levy *Curr Biol* 2015) which suggests it to be a potential alternative regulatory parameter.

3. The authors show exciting data on dissociation rates using their novel imaging approach. This is providing interesting insight into how the tagged TBP behaves in a biphasic manner. However, again the authors perhaps overinterpret the validity of their finding to endogenous TBP (see concerns above regarding the reagent used). Additionally to the issue of the reagent, old and recent observations argue for TBP to bind stably to DNA even during mitosis (Chen et al., 2002, DOI: 10.1091/mbc.01-10-0523; Oelgeschlager et al., 2002, DOI: 10.1038/ncb733 react-text: 72, Teves et al., 2017, DOI: 10.7554/eLife.22280). In case of such stable, likely non-stochastic binding by endogenous proteins, one wonders how exogenously provided TBP will mimic endogenous behaviour.

4. "Nuclear size regulates TBP chromatin associations"

This claim is not substantiated by the sets of evidence provided. Evidence is provided only for correlation, but causative effects can only be proven by interference experiments, such as direct manipulation of nuclear size (e.g. Jevtic and Levy, *Curr Biol* 2015).

Secondly, the assumption, that concentration of accessible TBP binding sites is constant before and during ZGA is solely based on the observation that Mnase digested nucleosome free regions are constant before and at MBT stages as identified by the cited work of Zhang et al., 2014. Firstly, the Mnase-seq approach used does not interrogate repeats, which is a large chunk of the genome, therefore the estimates are not representative for the whole genome, secondly the method used to compare early and late stages involved 10kb windows (Supplemental Fig 2). This is too large to identify nucleosome occupancy of promoters/enhancers (typically few hundred bp) with TBP binding sites. Moreover, even if it was true, 20% DNA is still hundreds of Mb representing up to a million units of potential binding sites. Taken together, the assumption of constant binding sites is a useful one for building simplifying theoretical models, but is far from certain to reflect true chromatin dynamics in the embryo. The previously demonstrated dramatic increase of occupancy of promoters by transcriptional activity/activation associated histone marks (e.g. H3K4me3, Lindeman et al., 2011 Vastenhouw et al., 2010), pre ZGA PolII binding and increasing concentration of pioneering factors (Onichthchouk et al., 2014, Lee et al., 2014) together suggest a dynamic change in the opening of chromatin, which is not in line with the proposed constant nucleosome free status of accessible sites.

The experiment with TSA is poorly described and lacks a clearly defined hypothesis. Presumably the authors suggest that an HDAC inhibitor will lead to more open chromatin in early embryos. The expected and observed effect of TSA on "DNA content" which one assumes is reference to "TBP-accessible DNA content" is not explained and not validated independently. No reference is provided to suggest how TSA affects chromatin in cleavage stage embryos.

How do we arrive to the estimated number of molecules (referring to bound? - again not clearly phrased) is only possible to decipher by reading the supplemental material and with some difficulty.

5. As a general comment, I found the narrative of the ms to be difficult to follow, it is unclear why are materials, which are critical to the understanding of the logic and narrative often deposited in supplemental material. This makes the flow and understanding of the key arguments laid down in the main text somewhat cumbersome. The authors often change terminologies between figures and text, which again makes the reading difficult (DNA content, accessible binding sites and bound molecules appear to be used interchangeably).

5. The authors argue that they have demonstrated “determination of the number of accessible DNA binding sites” for TBP. No such numbers appear in the main body of the manuscript except for presentation of arbitrary units on plots. There are bound molecule numbers cited in various parts of the text, but they are poorly described and it is unclear what they are supposed to mean, for example: “534612 molecules in 11 embryos”. How is this figure to be read?

6. Chromatin compaction during replication

This section is again cryptic, while it does provide interesting information on changing binding kinetics within a cell cycle. Key information about the change of nuclear size is downgraded to Supplemental material, while it remains integral to the main figures and critically important to consider for the argument.

However, there is no evidence provided on what time points in the cell cycle do the authors consider as early and late replication. “Early phase” and “late phase” of cell cycle are referred to in Figure S10 while the text talks about “pre-replication” and “post replication”. Presumably both sets of terminologies refer to interphase nuclei without information collected about replication timing directly. Since a late phase of the cell cycle is likely overlapping with prophase of mitosis, it is expected that chromatin will be more compacted. However, prophase is expected to occur after replication. Given that there is no evidence for measuring replication in this study, as a result, the claim in the subheading “Chromatin is compacted during replication” is unsubstantiated and overinterpretation of their observation.

7. The abstract makes additional dramatic claims, which are not sufficiently addressed in the ms, such as “nucleus size as timer of ZGA”. No data is provided on how nuclear size and TBP binding affect ZGA in the ms. Thus, the statement “Shrinking nucleus as driving force” remains an overinterpretation both for TBP dynamics and the ZGA (see my comment about lack of causative links). Same holds true for the term “enforces” in the title.

Minor comments

There is no page numbering, making the navigation in the ms and writing this review a bit more difficult.

The validation of the single molecule detection is not very well explained, and the statement “A sudden drop of fluorescence to the background level after several frames indicated the single molecule nature of fluorescence signal” needs further explanation in the main text for the non-specialist/developmental biologists who are target readers of this manuscript. This example is symptomatic about the way the ms is written and dampens the enthusiasm of those potential readers who are less trained in biophysics and single molecule imaging.

Reviewer #2:

Remarks to the Author:

Reisser and Gebhardt address the significant, but unresolved, question as to how the zygotic genome becomes activated during embryonic development. The authors use light sheet microscopy to measure the nuclear concentration of a key transcription factor (TBP) across the early stages of zebrafish embryos. From these measurements, they infer which fractions of TBP are specifically or non-specifically bound to DNA. These data were used to construct a model that relates these concentrations to the concomitant decrease in nuclear volume. One key conclusion is

that changes in nuclear size (i.e., volume) can influence the concentration of DNA and hence chromatin-transcription factor interactions, which presumably leads to altered levels of transcription.

This is a very nice study with several advances. First, the authors establish light sheet microscopy as an approach to visualize the spatial and temporal dynamics of a transcription factor in live zebrafish embryos. This allowed them to make direct measurements of TBP in single nuclei over time, which, as far as I know, is the first 'single-cell' measurement of transcription factor concentration in living zebrafish embryos. As much work in the field relies on genetics and genomics, the approach of single-molecule imaging should provide distinct and more quantitative insights in future studies. Second, this study draws attention to the role of nuclear volume on DNA concentration with respect to models of zygotic genome activation (ZGA). In current models that relate ZGA timing events to changes in the nuclear-to-cytoplasmic (N/C) ratio, the "N" is often assumed to be DNA ploidy, and effects of DNA concentration independent of ploidy have largely been ignored.

Overall, this manuscript makes an important contribution to our understanding of how transcription is activated during ZGA, and should be of interest to a broad audience interested in transcription, development, or nuclear organization. Some considerations and suggestions for authors follows:

Suggestions:

1) Can the authors discuss how their model fits with the results from the Levy group that show transcription increases in response to increased nuclear volume (Jevtic and Levy, *Current Biology*; 2015)? This reference is mentioned in the discussion ("Effect of nuclear size on the timing of ZGA has been shown."), but the Jevtic and Levy results are the opposite of what is predicted by the model presented here. The differences may simply be due to the different species used — frogs vs. fish — but the discrepancies should at least be noted and discussed.

2) Exogenous mRNA (mEos2-TBP) is used for the TBP reporter. To properly interpret the mEos2-TBP measurements, it is critical to understand whether mEos2-TBP translation is stable throughout early embryo development, and whether this effects translation of endogenous TBP. While complicated ways to assess this are likely outside the scope of the manuscript, it would be straightforward to assess bulk protein levels by performing a western blot for mEos2-TBP and endogenous TBP throughout the developmental time-course studied here. (Related—are the authors assuming a constant production of TBP? Or a stable pool of protein in large excess of what is required in the nucleus?)

Minor comments:

—Can the authors define "bound" and "chromatin-bound" more explicitly when referring to DNA-bound TBP, or just stick to a single usage for describing what I interpreted to mean "total" bound (i.e., specific plus non-specific bound)? I felt that "bound" and "chromatin-bound" were used inconsistently or interchangeably in the text and graphs.

--for those unfamiliar with the details of light-sheet microscopy: how does the signal-to-noise ratio presented here compare to what is usual? Is the SNR here particularly good? bad? standard?

—the assumption in the model that chromatin accessibility is unchanged across the time points analyzed here may not necessarily be valid. Recently published assays in mouse (ATAC-seq, DNase-seq) have suggested that chromatin accessibility is dynamic in the early embryo. Local changes in chromatin accessibility, unrelated to the cited unchanging 80% total nucleosome content, may affect specific TBP binding. Is there other data in zebrafish that can be cited to support this assumption?

—TSA is not explained in the text or methods. Can the authors provide information as to what this

reagent does to the embryo? Perhaps with a reference?

—Is there any relevance of the kink in the graph in 2a? It appears that there relative concentration of bound TBP increases dramatically at the MBT. Is it possible that 2 regimes exist, pre and post MBT?

--typo in the discussion "...not considered to the full 'extend'" should be 'extent'.

Reviewer #3:

Remarks to the Author:

Review of "Decreasing nuclear volume concentrates DNA and enforces transcription factor-chromatin associations during Zebrafish genome activation" by M. Reisser and J.C.M. Gebhardt.

In the following manuscript, Reisser and Gebhardt developed a new imaging technology and studied the dynamics of the transcription factor TBP during the crucial developmental period of zygotic genome activation in zebra fish embryo. The authors were able to image single molecules of TBP in the developing embryo using the reflected light-sheet microscope that the authors developed. By performing different modes of imaging, the authors were able to infer dynamics and the fraction of bound molecules as well as the residence time of the bound population. Combined with modeling of the nuclear volume as the embryo develops, the authors conclude that the decrease in nuclear volume is a key force in the timing of ZGA in zebrafish embryos.

The paper is technically sophisticated and performing single-molecule imaging in live zebrafish embryos is impressive. However, the paper is not written as a methods paper but as a "biological story" which provides a new model wherein "the shrinking nucleus is a major driving force and timer of ZGA in zebrafish embryos". As such, the paper should be evaluated on whether or not the data support the biological conclusions.

The model proposed by the authors is interesting and broadly seems plausible. However, the evidence to support the model is poor. The authors have a tendency to state a plausible hypothesis as a fact and then use successions of these to substantiate their model. However, many of these statements are not demonstrated and the whole paper reads as a bit of a house of cards. Accordingly, the authors must either severely tone down the title, text and discussion or provide sufficient evidence to support their conclusions.

The paper is also notable in that it basically involves one or two experiments described in minimal figures and then a very long discussion. If the authors want to demonstrate their model they should perform perturbation experiments and support the key conclusions with several orthogonal approaches.

Poorly supported statements

Authors state that "The increase in bound mEos2-TBP concentration was accompanied by an ~6-fold increase in the total nuclear concentration of mEos2-TBP molecules (Figure 2b and Methods). An equal rise of total and bound concentration would be expected from the law of mass action (Supplementary Figure S5)". We disagree. There are many potential non-linear effects, such as saturation of binding sites early. Another simple explanation would be that there is more transcription later. Or that there is more chromatin remodeling later. Also, TBP does not bind alone. What about the other TBP-binding partners – or is it really that TFIID is only expressed later and that TFIID facilitates TBP binding? What about RNA Pol II which interacts with TBP? Again the the model put forward by the authors, that it is due to a shrinking nucleus, is interesting and plausible. But it is one of many plausible models and the authors do no experiments to validate that their model is the right one. As such, their evidence is poor and too speculative to support a firm conclusion.

Authors say "To be able to draw conclusions also valid for endogenous TBP, we determined the fraction of chromatin-bound TBP molecules, a quantity independent of molecule concentration (Figure 2d and Methods)". Again, this is not true. The total bound fraction is not independent of

molecule concentration. In fact, over-expressing a protein affects the bound fraction and the residence time. For example, see the model of facilitated dissociation: <http://www.pnas.org/content/114/16/E3251> This is potentially troubling as the authors' experiments rely on microinjection of exogenous mEos-TBP RNA, with no indication as to how much is translated and therefore they have no idea of the concentration of their exogenous construct or the amount of TBP in the embryo.

And how does Figure 2d show that their conclusion is valid for endogenous TBP? How did they measure endogenous TBP? The figure legend for 2d does not mention endogenous TBP.

Authors state that "This result also holds for endogenous TBP, as the fraction of specifically bound molecules is independent of concentration." We disagree. The fraction of specifically bound molecules often change with over-expression as shown in many studies, so this statement is clearly wrong as documented by many examples. Second, authors have not shown any data on endogenous TBP and they have not demonstrated that mEos2-TBP is functional. It is perfectly possible for endogenous TBP to be more functional and bound to a higher extent than mEos2-TBP. Authors should not make strong statements when they do not have any evidence to support them.

Authors say that "Overall, the number of accessible DNA binding sites did not change significantly during replication (Figure 4b). This result suggests that DNA becomes compacted during replication such that sites accessible for TBP molecules approximately halve." It is not clear how the authors measured accessible sites for TBP. It seems that it is solely based on the single molecule imaging and fraction bound of TBP. This is not an appropriate measure for DNA accessibility. Moreover, the author say they did the second set of measurements "post-replication", so it should not matter what happened during replication since the measurement was taken after replication. Presumably they compare G1 to G2. It would be extremely surprising if the genome was 2-fold more compacted in G2 than in G1 and the authors should provide independent and compelling evidence if they want to make such a striking claim.

TBP residence time

The authors use time-lapse imaging (Gebhardt Nature Methods 2013) to measure the residence time. They find a residence time of ~7 seconds for TBP. This is surprisingly short given what has been previously published. Authors cite a yeast paper that reported a similar value, but other studies report 1-50 minutes (<https://www.nature.com/articles/srep39631>). In particular, Zebrafish is probably more similar to mammalian cells than yeast cells and previous FRAP on mammalian TBP found residence times on the order of minutes. It is concern that due to drift and photo-bleaching, the authors cannot measure stable (minute) binding events. Under the imaging conditions used by the authors, what is the photobleaching rate or half-life? The authors should state this number in the main text.

Also, the authors should either be explicit about the uncertainty of their residence time measurement or provide evidence for the same result with an orthogonal technique like FRAP.

Figures showing raw data

The authors show almost no raw data in the main figures or the supplement. Figure 2e is a good example. It nicely shows all the data in one place, but it is impossible to read the figure and they do not show legible figures in the supplement. They should show the same data in a log-log or spread out manner in the supplement. It is not possible to see what the frequency of ~40 s binding events were. Also given the relatively high density of molecules in the movies and the frequent case of molecules coming into focus from out of focus, it is not clear the authors can properly do the tracking when there are long dark times.

What do the numbers above the blue points mean in 2e? Is that the dark time? Too little detail is given.

Technical concerns

- Authors inject mEos2-TBP RNA molecules. What is their evidence that the mEos2-TBP fusion protein is fully functional? This is a major concern, because the fusion protein is expressed in a background where the wild-type protein is also expressed.

- Authors say mEos2-TBP concentration increases by about ~25 fold from 64-cell stage to oblong stage. How do the authors know that translation of their injected mRNA is physiologically regulated. The simplest model is that mRNA is constantly translated and the longer you wait, the more protein you get instead of this being a case of physiological regulation.
- There are a lot of concerns about the concentration measurements. And the authors do not explain in sufficient detail how they did them. They say they change the photoactivation period depending on the mEos2 concentration – this means that they compare different conditions? Also in a 0.5s vs. a 5s photoactivation period, non-photo-activated mEos molecules can move in and out of focus of the activation laser lightsheet to different extents giving rise to complicated relationships between pulse duration and extent of photo-activation.
- How much embryo-to-embryo variability in the amount of injected mRNA was there? How did authors control for this? How did authors ensure that mRNA was uniformly distributed to avoid high local concentration gradients? Diffusion constants of mRNAs are very low, so this could be a big problem.
- In single-molecule imaging it is critical to correct for photobleaching, which does not happen during the dark periods, and which is obviously important to correct for. But living embryos move a lot and molecules can also move out-of-focus and disappear further complicating such corrections. Drift, unlike photobleaching, should depend on time and thus increase with increasing dark time. The residence time fitting model ignores this and I think this might be why they claim to calculate such a short TBP residence time. Authors should correct for drift carefully.
- Why are the units for the number of accessible DNA sites (3b) in AU? It should be number per genome (e.g. 7000 per genome).

Reviewers' comments:

Reviewer #1:

This ms is an exciting report on the application of a novel single molecule imaging approach to detect TBP proteins in nuclei of developing zebrafish embryos. As a developmental biologist, I found the primary imaging data impressive which represents a breakthrough in real time in vivo imaging of transcription factor activity in living vertebrate embryos. The manuscript builds on the technology and demonstrates convincing correlation between binding of a misexpressed, tagged TBP to chromatin and dynamics of nucleus size during the cleavage stages prior to global genome activation.

We thank the reviewer for recognizing the importance and novelty of our work.

The observed binding profile suggests gradual increase of chromatin accessibility prior to ZGA and will be useful entry into revisiting the mechanisms of zygotic genome activation. However, in its current form the ms is hindered by overinterpretation of the findings and goes too far in concluding on the causative relationship between nucleus size and triggers of genome activation. The measurements are elegant, but remain correlative and do not provide sufficient lines of causative evidence (which one would be able to generate by interference experiments) to substantiate the dramatic claims presented in the title, abstract and subheadings of the Results section.

We realized that some of our interpretations, the causative relation between nuclear size and timing of ZGA, as well as the compaction of chromatin during replication, might have been preliminary or gone too far. Other claims such as a causative relation between decreasing nuclear size and increasing chromatin-bound fraction of TBP are valid but we feel that they were not well presented in the manuscript.

We now very substantially revised our manuscript. We added more controls, improved data analysis and added additional measurements on a second transcription factor, Sox19b. Most importantly, we extensively revised the presentation of our reasoning and clarified which of our measured parameters we use for quantitative analysis. Accordingly, we kept valid statements, discussed our interpretations more carefully and toned down overenthusiastic interpretations.

We recognized that our reasoning was not well presented and could be misunderstood. In particular, we now more clearly state the difference between numbers (or concentrations) of molecules and the percentage (or fraction) of bound molecules, which is independent of absolute numbers in absence of saturation. Moreover, a causative relation between nuclear size and bound fraction is not a possible model amongst many others to explain our data, but a consequence of the law of mass action of systems at equilibrium, and needs to be accounted for in any mechanism dealing with binding of transcription factors.

Besides these issues, there is a concern with the structure of the ms: lots of the novel technology and important sets of evidence are presented in the supplementary material for no obvious reason. The ms would be better suited to repitch as a primarily technology paper with elegant demonstration of utility by showing the TBP binding profiles and their correlation with nuclear size. The overemphasis on the interpretation of the underlying biology begs for additional experiments, and does not give justice to an otherwise excellent technological advance.

We agree that some of our novel developments were hidden in the supplements in the previous version of our manuscript. We now reorganized the manuscript to emphasize more our technical advances. At the same time, we kept valid statements but toned down more speculative interpretations.

Major points

1. Single molecule imaging and detection of chromatin binding by overexpressed TBP

The imaging data is generated by a novel RLSM imaging approach and processed by innovative imaging and image processing pipeline which led to impressive and convincing observations on the level of detection and quantification of the single molecule binding signals.

We are glad the reviewer judges our results as impressive and convincing.

What is questionable is whether the data truly reflects endogenous TBP functions.

Firstly, there is no information given about the key reagent (mEos2-TBP) of their study (apart from a statement that it has been a gift from a colleague and otherwise unpublished). This lack of information raises several questions:

As suggested, we now added more information on our fusion constructs (see below).

what is the origin (species) of the TBP used, how was the fusion with Eos2 designed,

We now name the Zebrafish origin of TBP and Sox19b in the text (second paragraph of results) and provide information on the sequence in the Methods.

how was the fusion validated (does the fusion TBP still bind DNA with similar efficiency to wild type,

We now included a Western Blot measurement of possible overexpression of mEos2-TBP compared to endogenous TBP. We found only marginal overexpression at the mRNA amount we injected (Figure 2c). Additionally, we now added ChIP-qPCR experiments for mEos2-TBP fused to an HA tag suggesting that mEos2-TBP binds to genes of endogenous TBP binding (Figure 2d).

We also added ChIP-qPCR experiments for Sox19b fused to HA tag demonstrating preferred binding to specific genes compared to a genomic control sequence (Figure 2e). A comparison to endogenous Sox19b binding could unfortunately not be performed due to lack of a good Sox19b antibody. This also prevented a test for overexpression, however, we used only half the injected amount of mRNA compared to mEos2-TBP. Measurements of the chromatin-bound fraction do not indicate that saturation would occur (Figure 4c and d). These experiments indicate that chromatin-binding of mEos2-TF constructs approximates binding of endogenous TF.

does it remain biochemically and biologically functional,

Our mEos2-TBP is closely related to a recently published CRISPR knock-in fusion protein of TBP in mammalian cells, which shows normal function (Teves et al., bioRxiv 2018) and previous publications using a FP-TBP fusion construct (e.g. Chen et al., 2002). It is thus very likely that also our mEos2-TBP fusion protein functions normally. The conclusions of our manuscript are based on chromatin binding and data suggests proper binding of mEos2-TFs (Figure 2d and e). Besides expression level and in vivo binding behaviour we also tested for effects on morphology and timing of development upon mRNA injection (Figure 2f)(see below).

We now added measurements of a second TF, Sox19b, which is biologically very different from TBP, and found a similar behaviour of its chromatin-bound fraction compatible with a decreasing nuclear size. While we recognize that some biological function might differ between the situation of added fusion proteins and wild type conditions, we do not believe such differences will affect our conclusions.

does its overexpression by microinjected mRNA affect development and timing of MBT,

As suggested, we performed a comparison of expression levels between fusion TBP construct and wild type TBP by Western Blot and found only marginal overexpression (Figure 2c). In addition, we tested whether development and timing of MBT was altered based on the morphology at different stages in early development (Figure 2f). For Sox19b, development was delayed after sphere stage. The developmental delay observed for embryos into which Sox19b mRNA was injected is comparable to previous observations with embryos into which Sox3, which is functionally related to Sox19b, was injected (Shih et al., 2010) (Figure 2f). Moreover, our new analysis relating the bound fraction of TFs to all detected TFs indicates that we do not have saturation of binding sites (Figure 4c and d).

does it affect the measured cellular and cell cycle parameters (size, cycle length etc.)

As suggested, we checked that cycle length did not change between mEos2-TBP and mEos2-Sox19b injected embryos and wild type embryos up to sphere stage (Figure 2f).

We now also found that nuclear size did not differ between embryos with mEos2-TF mRNA injected and wild type embryos (Supplementary Figure S8).

and how does its distribution and concentration compare to endogenous TBP?

As suggested by the reviewer and as detailed above, we now provide a Western Blot showing that mEos2-TBP is only marginally overexpressed compared to endogenous TBP. The distribution of mEos2-TBP was almost entirely nuclear, as expected from endogenous TBP (see supplementary movie 2).

We note that our study is designed such that concentrations or changes in concentration cancel out, since we focus on and analyse the chromatin-bound fraction of TFs (number of bound TFs divided by all detected TFs). Thus, potential deviations in the regulation of mEos2-TF and endogenous TF expression level do not affect our measurements or conclusions.

The reader is left to assume without experimental evidence whether the detected fusion molecule is representative for the distribution and behaviour of TBP proteins in early embryos.

As suggested by the reviewer and as detailed above, we now added several experiments validating a reasonable expression level of fusion protein, binding to genes to which endogenous protein binds and negligible effects on morphology or timing during the time period covered by our measurements.

Secondly, TBP is known to be highly dynamic in frogs and fish (publications by Veenstra and Muller labs), it is deposited maternally alongside 2 other TBP family members with distinct dynamics, but partially overlapping functions and post-transcriptionally regulated to dramatically increase protein levels just before MBT (Veenstra, Woolfe). The concentration of exogenous TBP after mRNA injection is unlikely to follow this pattern.

We agree with the reviewer that expression of our exogenous proteins will not follow endogenous regulation and might therefore exhibit different concentrations. We therefore designed our study such that these differences do not affect our analysis or conclusions, by focusing on the chromatin-bound fraction of molecules. This was not well motivated in our previous manuscript. We now state this fact much more clearly (“Since the number of detected mEos2-TF molecules does not report on endogenous TF concentrations, we did not directly evaluate measured protein concentrations.”) and refrain from reporting concentrations in the main text, to prevent possible confusions.

Nevertheless, our methodology allows accurate measurements of the concentration of fluorescent fusion proteins. This will become important if for example CRISPR knock-ins become more generally available. We thus still report the concentration measurements in the supplements.

Thirdly, there seems to be albeit, not huge but consistent discrepancy between their model and measured TBP binding between 64 and 128 cell stages (Fig 3a and Fig4a) is this difference due to a factor that is not modelled by injected mRNAs excluded from posttranscriptional regulation (?).

Posttranscriptional regulation affects the expression level of a protein. As discussed above, our measurement design is not susceptible to expression levels.

However, regarding the trend of data in those figures, the reviewer mentions an important point, which is related to a point also addressed by reviewer 2 and 3. We initially applied two different measurement protocols. In one, the power of the activation laser was constant over development, but in the other this power was not controlled, which added a systematic detection noise to our measurement. We now restricted our analysis to the condition with controlled activation power to avoid this systematic instrumental noise.

Such regulatory information is likely carried in the non-coding sequences of TBP, are these sequences present on the expression construct? Since we are not given information on the reporter expression construct it is unclear how much does this construct mimic endogenous TBP and its translational regulation.

The expression constructs of mEos2-TBP and mEos2-Sox19b did not include non-coding sequences. However, as discussed above, such regulatory sequences affect expression levels, which are not of importance to our experimental design in the limit of small overexpression.

The supplemental AVI files, show cytoplasmic foci similar to those in the nucleus. Can these be explained?

The cytoplasmic fluorescent signals are due to mEos2-TF fusion proteins that are not yet transported into the nucleus. Both mEos2-TFs localized almost exclusively to the nucleus. However, with single molecule sensitivity, even trace amounts in the cytoplasm can be detected. We found that no such spots are visible without injection of mRNA encoding for mEos2-TBP (Supplementary Figure S2).

2. Why do the authors ignore the nucleo-cytoplasmic ratio as parameter in their study and why do they suggest nuclear volume (without cytoplasm/cell volume) as an independent factor defining TBP concentration dynamics? Since the NC ratio is a well-known and very well established regulator of zygotic genome activation, and likely regulating chromatin accessibility (by for example the dilution of the repressive factor such as histones, see works by the Gurdon and Vastenhouw labs) one would expect that besides nuclear volume, cellular volume measurements are also carried out together with TBP binding to test if the fit is better with NC ratio than by nuclear volume only.

As pointed out by the reviewer, the nucleo-cytoplasmic volume ratio is indeed an important parameter. In particular, as mentioned by the referee, it might affect the absolute number of a species able to pass the nuclear pore (Kopito et al., PNAS 2007) such as a TF or a repressor of TF binding (Newport et al., Cell 1982) and thereby might affect nuclear protein concentrations and particularly the apparent number of accessible binding sites. We now mention this possible influence of the nucleo-cytoplasmic volume ratio in the introduction and added a paragraph to the discussion section.

As suggested by the referee, we now have included a possible effect of the nucleo-cytoplasmic volume ratio on the apparent number of chromatin binding sites in our analysis. Based on the

suggestion of this reviewer in comment 4, we now dropped the unsubstantiated assumption of constant DNA accessibility. Instead, we used this parameter as a read out in every stage when applying the physical model of TF-chromatin interactions to our data. This allowed us to observe a trend of DNA accessibility over development (Figure 5d). For mEos2-TBP, we observed an increase in apparent number of chromatin binding sites, while it was constant for mEos2-Sox19b (see also below).

Cell volumes are heterogeneous and change non-linearly between cell cycles (Kane and Kimmel, Development 1993). Moreover, the NC ratio dynamic appears to change dramatically in early Xenopus development (Jevtic and Levy Curr Biol 2015) which suggests it to be a potential alternative regulatory parameter.

As detailed above, we agree that nucleo-cytoplasmic volume ratio is an important parameter and we have included a possible effect on the apparent number of chromatin binding sites in our analysis.

3. The authors show exciting data on dissociation rates using their novel imaging approach. This is providing interesting insight into how the tagged TBP behaves in a biphasic manner. However, again the authors perhaps overinterpret the validity of their finding to endogenous TBP (see concerns above regarding the reagent used). Additionally to the issue of the reagent, old and recent observations argue for TBP to bind stably to DNA even during mitosis (Chen et al., 2002, DOI: 10.1091/mbc.01-10-0523; Oelgeschlager et al., 2002, DOI: 10.1038/ncb733 react-text: 72, Teves et al., 2017, DOI: 10.7554/eLife.22280). In case of such stable, likely non-stochastic binding by endogenous proteins, one wonders how exogenously provided TBP will mimic endogenous behaviour.

As stated above, we now included more information about our constructs and new measurements suggesting sound binding of our fusion proteins.

Regarding the residence times we measured, we agree that our measurements differ to some publications, but not all. The discrepancy likely arises due to differences in TBP binding between species or due to fast cell cycles in early Zebrafish embryos. We note that ChIP experiments by Oelgeschlager et al. 2002 do not measure a residence time but freeze a current state. Other publications using fusion proteins do not measure the behaviour of endogenous protein, similar to our study. Fast TBP cycling is compatible with measurements of RNA Polymerase II dynamics in live cells that suggest fast transcription initiation times of ~8 s (Cho et al. 2016).

Our measurements do not provide a hint for minute-long binding of molecules in interphase cells. However, we cannot completely exclude a potential contribution of drift and out-of-focus movement of bound molecules during dark times to the observed residence time, which might lead to a potential underestimation of our residence time.

We added a corresponding discussion to the results section.

4. "Nuclear size regulates TBP chromatin associations"

This claim is not substantiated by the sets of evidence provided. Evidence is provided only for correlation, but causative effects can only be proven by interference experiments, such as direct manipulation of nuclear size (e.g Jevtic and Levy, Curr Biol 2015).

We realized that our reasoning was not well presented in the previous manuscript. It is important to note that a causative relation between nuclear size and bound fraction is a direct consequence of the law of mass action of systems at equilibrium. According to this physico-chemical law, the chromatin-bound fraction depends on the concentration of TF binding sites and on the rate constants of association and dissociation of the TF. Binding site concentration in turn is determined by dividing the number of binding sites accessible to this TF on chromatin by the volume of the nucleus. Overall, any physical or biological factor will either directly or indirectly affect one of the parameters nuclear size,

kinetic rate constants or number of accessible binding sites.

As an example, as the reviewer suggested, a change in nucleo-cytoplasmic volume ratio might be reflected by a change in the number of accessible binding sites for a TF.

Consequently, a causative relation between nuclear size and bound fraction is not just one possible model amongst many others to explain our data. This relation rather needs to be accounted for in any mechanism dealing with binding of transcription factors and adds to other possible effects, e.g. changes in DNA accessibility.

We now added a figure (Figure 1b) and several paragraphs to the introduction to describe this point much more clearly, and reflected on possible influences on the bound fraction.

As also suggested by reviewer 3, it is important to test whether the law of mass action in its simple form is valid here, or whether further dependencies, such as a concentration-dependent dissociation rate constant, might come into play. We thus added additional analysis (Figure 4c and d and Supplementary Figure S6) to test for this. From the relation between bound molecules and all detected molecules in every stage (Figure 4c and d and Supplementary Figure S6), we assume that the law of mass action is valid in every stage. Consequently, we take the effect of shrinking nuclear volume on the chromatin-bound fraction of a TF as given.

Nevertheless, we tested whether we could artificially decrease nuclear size in Zebrafish embryos, analogous to the experiments by Jevtic et al., 2015 (please see Additional Figure A1). Interestingly, upon injection of Zebrafish Rtn4a in addition to mEos2-TBP, we did not observe significant changes to nuclear size, however huge changes to the chromatin-bound fraction of mEos2-TBP, along with huge changes in DNA accessibility. The effect of Rtn4a, at least in our case, thus seems to be rather in changing chromatin structure than nuclear size. We thus do not think that injection of Rtn4a, and probably other means of increasing nuclear size, will be a good control experiment, since not only nuclear size but other, unrecognized changes are likely to occur. We thus refrained from further artificially changing nuclear size in the live Zebrafish embryo.

If the reviewers feel that adding these experiments, which we think demonstrate that injection of reticulon 4a to Zebrafish embryos is not a good way to change nuclear size without affecting other cellular parameters, will improve the manuscript, we will be happy to include these experiments.

Secondly, the assumption, that concentration of accessible TBP binding sites is constant before and during ZGA is solely based on the observation that Mnase digested nucleosome free regions are constant before and at MBT stages as identified by the cited work of Zhang et al., 2014. Firstly, the Mnase-seq approach used does not interrogate repeats, which is a large chunk of the genome, therefore the estimates are not representative for the whole genome, secondly the method used to compare early and late stages involved 10kb windows (Supplemental Fig 2). This is too large to identify nucleosome occupancy of promoters/enhancers (typically few hundred bp) with TBP binding sites. Moreover, even if it was true, 20% DNA is still hundreds of Mb representing up to a million units of potential binding sites. Taken together, the assumption of constant binding sites is a useful one for building simplifying theoretical models, but is far from certain to reflect true chromatin dynamics in the embryo. The previously demonstrated dramatic increase of occupancy of promoters by transcriptional activity/activation associated histone marks (e.g. H3K4me3, Lindeman et al., 2011 Vastenhouw et al., 2010), pre ZGA PolIII binding and increasing concentration of pioneering factors (Onichthchouk et al., 2014, Lee et al., 2014) together suggest a dynamic change in the opening of chromatin, which is not in line with the proposed constant nucleosome free status of accessible sites.

We thank the reviewer for pointing out the various issues with our assumption and agree that in light of these points it does not seem to be appropriate. As suggested, we now do not assume a constant

DNA accessibility any more. Instead, we extract the apparent number of chromatin binding sites as free parameter from our data. To do so we insert the measured parameters bound fraction, nuclear size and dissociation rate constant into the law of mass action model of TF-chromatin binding (Figure 5d). In this approach, the apparent number of chromatin binding sites includes all effects, which might alter DNA accessibility of a TF such as an opening of chromatin or titration of binding inhibitors, but also possible changes in association rate constant, which we cannot measure in vivo.

We feel that this new approach to analyse our data is indeed better suited to deal with the complexity of the developing embryo. It allows us to look at the different factors affecting TF-chromatin interactions in a more differentiated way.

For mEos2-TBP, the data indicate that both nuclear size and apparent number of chromatin binding sites contribute to an increase in the bound fraction. In contrast, for mEos2-Sox19b, data indicate that nuclear size predominantly contributes to this increase.

The experiment with TSA is poorly described and lacks a clearly defined hypothesis. Presumably the authors suggest that an HDAC inhibitor will lead to more open chromatin in early embryos. The expected and observed effect of TSA on "DNA content" which one assumes is reference to "TBP-accessible DNA content" is not explained and not validated independently. No reference is provided to suggest how TSA affects chromatin in cleavage stage embryos.

How do we arrive to the estimated number of molecules (referring to bound? - again not clearly phrased) is only possible to decipher by reading the supplemental material and with some difficulty.

We agree that this experiment was not well motivated and described. Since it does not add any value to the current manuscript, we now omit it.

5. As a general comment, I found the narrative of the ms to be difficult to follow, it is unclear why are materials, which are critical to the understanding of the logic and narrative often deposited in supplemental material. This makes the flow and understanding of the key arguments laid down in the main text somewhat cumbersome. The authors often change terminologies between figures and text, which again makes the reading difficult (DNA content, accessible binding sites and bound molecules appear to be used interchangeably).

We agree that the presentation of our manuscript could be improved. As suggested, we now extensively rearranged the manuscript. In particular, we moved material from the supplements to the main text (e.g. the description of the novel illumination scheme), we more clearly explain our reasoning, included more background information on materials and harmonized terminologies.

5. The authors argue that they have demonstrated "determination of the number of accessible DNA binding sites" for TBP. No such numbers appear in the main body of the manuscript except for presentation of arbitrary units on plots. There are bound molecule numbers cited in various parts of the text, but they are poorly described and it is unclear what they are supposed to mean, for example: "534612 molecules in 11 embryos". How is this figure to be read?

There is currently no possibility to calibrate the value for the apparent number of chromatin binding sites. In our previous manuscript, DNA content (which we now renamed as suggested by the reviewer) was given relative to embryos injected with mRNA for mEos2-TBP. In our revised manuscript, we report the apparent number of chromatin binding sites normalized to the 64-cell stage of the respective injection condition.

The number of molecules and embryos mentioned by the reviewer refer to the number of molecules detected in our measurements and entering the analysis, and the number of embryos used for these

measurements. We now more clearly describe these numbers.

6. Chromatin compaction during replication

This section is again cryptic, while it does provide interesting information on changing binding kinetics within a cell cycle. Key information about the change of nuclear size is downgraded to Supplemental material, while it remains integral to the main figures and critically important to consider for the argument.

However, there is no evidence provided on what time points in the cell cycle do the authors consider as early and late replication. “Early phase” and “late phase” of cell cycle are referred to in Figure S10 while the text talks about “pre-replication” and “post replication”. Presumably both sets of terminologies refer to interphase nuclei without information collected about replication timing directly. Since a late phase of the cell cycle is likely overlapping with prophase of mitosis, it is expected that chromatin will be more compacted. However, prophase is expected to occur after replication. Given that there is no evidence for measuring replication in this study, as a result, the claim in the subheading “Chromatin is compacted during replication” is unsubstantiated and overinterpretation of their observation.

We agree that we did not describe this analysis very well. Since it does not add any value to the current manuscript, we now omit it.

7. The abstract makes additional dramatic claims, which are not sufficiently addressed in the ms, such as “nucleus size as timer of ZGA”. No data is provided on how nuclear size and TBP binding affect ZGA in the ms. Thus, the statement “Shrinking nucleus as driving force” remains an overinterpretation both for TBP dynamics and the ZGA (see my comment about lack of causative links).

Indeed, the phrase “the shrinking nucleus is a major driving force and timer of ZGA...” has maybe been too enthusiastic. We now rephrase it to better emphasize our technological advance: “Our single molecule approach provides quantitative insight into changes of TF-chromatin associations during the developmental period embracing ZGA.”

Regarding TF binding, we again note that the effect of nuclear size on the chromatin-bound fraction of a TF is a consequence of the law of mass action. Thus, calling a shrinking nucleus a driving force of TF binding is a valid statement. We now much more clearly differentiate different contributions to the law of mass action.

Same holds true for the term “enforces” in the title.

As above, our title “Decreasing nuclear volume concentrates DNA and enforces transcription factor–chromatin associations...” contains a valid statement. However, we now rephrase it to “Decreasing nuclear volume concentrates DNA and facilitates transcription factor–chromatin associations...”.

Minor comments

There is no page numbering, making the navigation in the ms and writing this review a bit more difficult.

As suggested, we now added page numbers to the manuscript.

The validation of the single molecule detection is not very well explained, and the statement “ A sudden drop of fluorescence to the background level after several frames indicated the single molecule nature of fluorescence signal” needs further explanation in the main text for the non-specialist/developmental biologists who are target readers of this manuscript. This example is symptomatic about the way the ms is written and dampens the enthusiasm of those potential readers

who are less trained in biophysics and single molecule imaging.

We apologize if our manuscript was not always written in a very clear way. We now add a reference to this common criterion of single molecule detection. Furthermore, throughout the manuscript, we tried to better explain our reasoning and methodological approaches to make the manuscript as clear as possible.

Reviewer #2:

Reisser and Gebhardt address the significant, but unresolved, question as to how the zygotic genome becomes activated during embryonic development. The authors use light sheet microscopy to measure the nuclear concentration of a key transcription factor (TBP) across the early stages of zebrafish embryos. From these measurements, they infer which fractions of TBP are specifically or non-specifically bound to DNA. These data were used to construct a model that relates these concentrations to the concomitant decrease in nuclear volume. One key conclusion is that changes in nuclear size (i.e., volume) can influence the concentration of DNA and hence chromatin-transcription factor interactions, which presumably leads to altered levels of transcription.

This is a very nice study with several advances. First, the authors establish light sheet microscopy as an approach to visualize the spatial and temporal dynamics of a transcription factor in live zebrafish embryos. This allowed them to make direct measurements of TBP in single nuclei over time, which, as far as I know, is the first 'single-cell' measurement of transcription factor concentration in living zebrafish embryos. As much work in the field relies on genetics and genomics, the approach of single-molecule imaging should provide distinct and more quantitative insights in future studies. Second, this study draws attention to the role of nuclear volume on DNA concentration with respect to models of zygotic genome activation (ZGA). In current models that relate ZGA timing events to changes in the nuclear-to-cytoplasmic (N/C) ratio, the "N" is often assumed to be DNA ploidy, and effects of DNA concentration independent of ploidy have largely been ignored.

Overall, this manuscript makes an important contribution to our understanding of how transcription is activated during ZGA, and should be of interest to a broad audience interested in transcription, development, or nuclear organization. Some considerations and suggestions for authors follows:

We thank the reviewer for this positive judgement of our work.

We realized that some of our interpretations, the causative relation between nuclear size and the timing of ZGA, as well as the compaction of chromatin during replication, might have been preliminary or gone too far. Other claims such as a causative relation between decreasing nuclear size and increasing bound fraction of TBP are valid but we feel that they were not well presented in the manuscript.

We now very substantially revised our manuscript. We added more controls, improved data analysis and added additional measurements on a second transcription factor, Sox19b. Most importantly, we extensively revised the presentation of our reasoning and clarified which of our measured parameters we use for quantitative analysis. Accordingly, we kept valid statements, discussed our interpretations more carefully and toned down overenthusiastic interpretations.

Suggestions:

1) Can the authors discuss how their model fits with the results from the Levy group that show transcription increases in response to increased nuclear volume (Jevtic and Levy, Current Biology;

2015)? This reference is mentioned in the discussion (“Effect of nuclear size on the timing of ZGA has been shown.”), but the Jevtic and Levy results are the opposite of what is predicted by the model presented here. The differences may simply be due to the different species used — frogs vs. fish — but the discrepancies should at least be noted and discussed.

As suggested, we now discuss this publication in the discussion section:

“...Jevtic et al. artificially increased or decreased nuclear size by adding reticulon or other factors and observed premature activation or delay of zygotic transcription when nuclear size was increased or decreased (Jevtic et al., 2015). These experiments do not contradict our measurements that revealed an increased chromatin-bound fraction of TFs when nuclei were small. Increasing the nucleocytoplasmic volume ratio of individual cells as done by Jevtic et al. might change the concentrations of a TF or a repressor of TF binding (Kopito et al., 2007; Newport et al., 1982), or otherwise change chromatin structure compared to an unchanged reference cell of the same stage. In effect, the apparent number of chromatin binding sites and the absolute number of bound TFs might be increased, leading to premature transcription.”

We have extensively rewritten and reorganized the manuscript, to increase the clarity of our reasoning. In addition, we tested whether we could artificially decrease nuclear size in Zebrafish embryos, analogous to the experiments by Jevtic et al., 2015 (please see Additional Figure A1). Interestingly, upon injection of Zebrafish Rtn4a in addition to mEos2-TBP, we did not observe significant changes to nuclear size, however huge changes to the chromatin-bound fraction of mEos2-TBP, along with huge changes in DNA accessibility. The effect of Rtn4a, at least in our case, thus seems to be rather in changing chromatin structure than nuclear size. We thus do not think that injection of Rtn4a, and probably other means of increasing nuclear size, will be a good experiment, since not only nuclear size but other, unrecognized changes are likely to occur. We thus refrained from further artificially changing nuclear size in the live Zebrafish embryo.

If the reviewers feel that adding these experiments, which we think demonstrate that injection of reticulon 4a to Zebrafish embryos is not a good way to change nuclear size without affecting other cellular parameters, will improve the manuscript, we will be happy to include these experiments.

2) Exogenous mRNA (mEos2-TBP) is used for the TBP reporter. To properly interpret the mEos2-TBP measurements, it is critical to understand whether mEos2-TBP translation is stable throughout early embryo development, and whether this effects translation of endogenous TBP. While complicated ways to assess this are likely outside the scope of the manuscript, it would be straightforward to assess bulk protein levels by performing a western blot for mEos2-TBP and endogenous TBP throughout the developmental time-course studied here. (Related—are the authors assuming a constant production of TBP? Or a stable pool of protein in large excess of what is required in the nucleus?)

We agree with the reviewer that our exogenous mEos2-TBP might not follow the expression profile of endogenous TBP. We therefore designed our study such that these differences do not affect our analysis or conclusions, by focusing on the chromatin-bound fraction of molecules that is independent of concentration in the absence of saturation. This was not well motivated in our previous manuscript. We now state this fact much more clearly (“Since the number of detected mEos2-TF molecules does not report on endogenous TF concentrations, we did not directly evaluate measured protein concentrations.”) and refrain from reporting concentrations in the main text, to prevent possible confusions.

The time course of TBP expression in early Zebrafish development has been reported (Bartfai et al., 2004). We observed via Western Blot that the expression level of mEos2-TBP is close to endogenous

TBP levels in the 1k-cell stage (Figure 2c). In addition, we provide other experimental tests of our mEos2-TBP construct. We demonstrate via ChIP-qPCR that mEos2-TBP binds to genes to which also endogenous TBP binds (Figure 2d). Second, we tested that injection of mRNA for mEos2-TBP does not alter morphology and time of stages up to 24 hpf (Figure 2f). Third, we now added additional analysis that indicate that we do not have saturation of TBP binding sites (Figure 4c). Similar tests were performed for the newly added mEos2-Sox19b construct.

Although we do not report the concentration measurements in the main text any more, our methodology allows accurate measurements of the concentration of fluorescent fusion proteins. This will become important if for example CRISPR knock-ins become more generally available. We thus still report the concentration measurements in the supplements.

Minor comments:

—Can the authors define “bound” and “chromatin-bound” more explicitly when referring to DNA-bound TBP, or just stick to a single usage for describing what I interpreted to mean “total” bound (i.e., specific plus non-specific bound)? I felt that “bound” and “chromatin-bound” were used inconsistently or interchangeably in the text and graphs.

As suggested, we now harmonized our terminology. We indeed meant total binding and did not intend to differentiate between specific and unspecific binding. This is justified since the stable bound proportion of chromatin-bound molecules is approx. constant over early development (Figure 5b).

--for those unfamiliar with the details of light-sheet microscopy: how does the signal-to-noise ratio presented here compare to what is usual? Is the SNR here particularly good? bad? standard?

As suggested, we now compare the SNR to measurements in cells. SNR of 5 is commonly observed, SNR of 2-3 approaches the limit of < 2 of reasonable detection. We add this to the text.

—the assumption in the model that chromatin accessibility is unchanged across the time points analyzed here may not necessarily be valid. Recently published assays in mouse (ATAC-seq, DNase-seq) have suggested that chromatin accessibility is dynamic in the early embryo. Local changes in chromatin accessibility, unrelated to the cited unchanging 80% total nucleosome content, may affect specific TBP binding. Is there other data in zebrafish that can be cited to support this assumption?

The reviewer addresses a point related to comments of reviewer 1 and 3. As suggested, we now do not assume a constant DNA accessibility any more. Instead, we extract the apparent number of chromatin binding sites as free parameter from our data. To do so we insert the measured parameters bound fraction, nuclear size and dissociation rate constant into the law of mass action model of TF-chromatin binding (Figure 5d). In this approach, the apparent number of chromatin binding sites includes all effects, which might alter DNA accessibility of a TF such as an opening of chromatin or titration of binding inhibitors, but also possible changes in association rate constant, which we cannot measure in vivo.

We feel that this new approach to analyse our data is indeed better suited to deal with the complexity of the developing embryo. It allows us to look at the different factors affecting TF-chromatin interactions in a more differentiated way.

For mEos2-TBP, the data indicate that both nuclear size and apparent number of chromatin binding sites contribute to an increase in the bound fraction. In contrast, for mEos2-Sox19b, data indicate that nuclear size predominantly contributes to this increase.

—TSA is not explained in the text or methods. Can the authors provide information as to what this reagent does to the embryo? Perhaps with a reference?

We agree that this experiment was not well motivated and described. Since it does not add any value to the current manuscript, we now omit it.

—Is there any relevance of the kink in the graph in 2a? It appears that there relative concentration of bound TBP increases dramatically at the MBT. Is it possible that 2 regimes exist, pre and post MBT?

The reviewer mentions an important point, which is related to a point also addressed by reviewers 1 and 3. We initially applied two different measurement protocols. In one, the power of the activation laser was constant over development, but in the other this power was not controlled, which added a systematic detection noise to our measurement. We now restricted our analysis to the condition with controlled activation power to avoid this systematic instrumental noise.

In addition, we have considerably improved the analysis of the chromatin-bound fraction (Figure 5a), based on the suggestion of reviewers. As suggested, we now dropped the unsubstantiated assumption of constant DNA accessibility. Instead, we used this parameter as a read out in every stage when applying the physical model of TF-chromatin interactions to our data. This allowed us to observe a trend of the apparent number of chromatin binding sites over development (Figure 5d).

--typo in the discussion "...not considered to the full 'extend'" should be 'extent'.

We have corrected this typo.

Reviewer #3:

Review of "Decreasing nuclear volume concentrates DNA and enforces transcription factor-chromatin associations during Zebrafish genome activation" by M. Reisser and J.C.M. Gebhardt.

In the following manuscript, Reisser and Gebhardt developed a new imaging technology and studied the dynamics of the transcription factor TBP during the crucial developmental period of zygotic genome activation in zebra fish embryo. The authors were able to image single molecules of TBP in the developing embryo using the reflected light-sheet microscope that the authors developed. By performing different modes of imaging, the authors were able to infer dynamics and the fraction of bound molecules as well as the residence time of the bound population. Combined with modeling of the nuclear volume as the embryo develops, the authors conclude that the decrease in nuclear volume is a key force in the timing of ZGA in zebrafish embryos.

The paper is technically sophisticated and performing single-molecule imaging in live zebrafish embryos is impressive.

We thank the reviewer for this judgement of our work.

However, the paper is not written as a methods paper but as a "biological story" which provides a new model wherein "the shrinking nucleus is a major driving force and timer of ZGA in zebrafish embryos". As such, the paper should be evaluated on whether or not the data support the biological conclusions.

We realized that some of our interpretations, the causative relation between nuclear size and the

timing of ZGA, as well as the compaction of chromatin during replication, might have been preliminary or gone too far. Other claims such as a causative relation between decreasing nuclear size and increasing bound fraction of TBP are valid but we feel that they were not well presented in the manuscript.

We now very substantially revised our manuscript. We added more controls, improved data analysis and added additional measurements on a second transcription factor, Sox19b. Most importantly, we extensively revised the presentation of our reasoning and clarified which of our measured parameters we use for quantitative analysis. Accordingly, we kept valid statements, discussed our interpretations more carefully and toned down overenthusiastic interpretations.

The model proposed by the authors is interesting and broadly seems plausible. However, the evidence to support the model is poor. The authors have a tendency to state a plausible hypothesis as a fact and then use successions of these to substantiate their model. However, many of these statements are not demonstrated and the whole paper reads as a bit of a house of cards. Accordingly, the authors must either severely tone down the title, text and discussion or provide sufficient evidence to support their conclusions.

We recognized that our reasoning was not well presented and could be misunderstood. In particular, a causative relation between nuclear size and bound fraction is not a possible model amongst many others to explain our data, but a consequence of the law of mass action of systems at equilibrium, and needs to be accounted for in any mechanism dealing with binding of transcription factors. We detail this further below. As suggested by the reviewer, we now test whether the law of mass action in its simple form is valid here, or whether further dependencies, such as a concentration-dependent dissociation rate constant, might come into play (see below).

The paper is also notable in that it basically involves one or two experiments described in minimal figures and then a very long discussion.

We now have extensively revised the manuscript, which also solved this issue.

If the authors want to demonstrate their model they should perform perturbation experiments and support the key conclusions with several orthogonal approaches.

We now much more clearly describe our reasoning. In addition, we added numerous new experiments to substantiate our claims, or toned down our statements where necessary.

Poorly supported statements

Authors state that “The increase in bound mEos2-TBP concentration was accompanied by an ~6-fold increase in the total nuclear concentration of mEos2-TBP molecules (Figure 2b and Methods). An equal rise of total and bound concentration would be expected from the law of mass action (Supplementary Figure S5).”. We disagree. There are many potential non-linear effects, such as saturation of binding sites early.

We agree that the presentation of our reasoning in the previous manuscript was not well done and we thus extensively revised it. In particular, we now more clearly discuss the law of mass action and its various dependencies in the introduction. According to this physico-chemical law, the chromatin-bound fraction depends on the concentration of TF binding sites and by the rate constants of association and dissociation of the TF. Binding site concentration in turn is determined by dividing the number of binding sites accessible to this TF on chromatin on the volume of the nucleus. Overall, any physical or biological factor will either directly or indirectly affect one of the parameters nuclear size, kinetic rate constants or number of accessible binding sites.

As an example, as the reviewer suggested, there could be saturation effects, affecting the number of accessible binding sites, or a concentration-dependent off-rate, affecting the kinetic dissociation rate constant. We now added a figure (Figure 1b) and several paragraphs to the introduction to describe this point much more clearly, and reflected on possible influences on the chromatin-bound fraction.

Regarding the question of saturation, we performed a Western Blot to test for a possible overexpression of mEos2-TBP and found mEos2-TBP was only marginally overexpressed for the amount of 60 pg of mRNA injected (Figure 2c). For mEos2-Sox19b we could not perform a Western Blot due to lack of high quality Sox19b antibodies, however we injected only 29 pg of mRNA and analysed our data in every stage for possible binding site saturation, which we did not observe (Figure 4c and d and Supplementary Figure S6).

Another simple explanation would be that there is more transcription later.

We agree with the reviewer that absolute concentration of endogenous TBP is a point we cannot access, since our exogenous mEos2-TBP might not follow the expression profile of endogenous TBP. We therefore designed our study such that these differences do not affect our analysis or conclusions, by focusing on the chromatin-bound fraction of molecules, which is independent of absolute numbers in absence of saturation. As result, even if transcription or translation change during development, the bound fraction is a good parameter to look at. This was not well motivated in our previous manuscript. We now state this fact much more clearly (“Since the number of detected mEos2-TF molecules does not report on endogenous TF concentrations, we did not directly evaluate measured protein concentrations. In contrast, the bound fraction of mEos2-TF molecules is independent of concentration and therefore may reflect the behaviour of endogenous TFs independent of differences in expression.”) and refrain from reporting concentrations in the main text, to prevent possible confusions.

Nevertheless, our methodology allows accurate measurements of the concentration of fluorescent fusion proteins. This will become important if for example CRISPR knock-ins become more generally available. We thus still report the concentration measurements in the supplements.

Or that there is more chromatin remodeling later.

Changes in chromatin remodelling might affect accessibility of DNA for a TF. As with absolute concentrations, accessibility of DNA is a parameter we cannot directly measure. Based on comments by all reviewers, we now dropped our simplifying assumption of constant DNA accessibility during development. Instead, we extract the apparent number of chromatin binding sites as free parameter from our data. To do so we insert the measured parameters bound fraction, nuclear size and dissociation rate constant into the law of mass action model of TF-chromatin binding (Figure 5d). In this approach, the apparent number of chromatin binding sites includes all effects, which might alter DNA accessibility of a TF such as an opening of chromatin or titration of binding inhibitors, but also possible changes in association rate constant, which we cannot measure in vivo.

We feel that this new approach to analyse our data is indeed better suited to deal with the complexity of the developing embryo. It allows us to look at the different factors affecting TF-chromatin interactions in a more differentiated way.

For mEos2-TBP, the data indicate that both nuclear size and apparent number of chromatin binding sites contribute to an increase in the bound fraction. In contrast, for mEos2-Sox19b, data indicate that nuclear size predominantly contributes to this increase.

Also, TBP does not bind alone. What about the other TBP-binding partners – or is it really that TFIID is only expressed later and that TFIID facilitates TBP binding? What about RNA Pol II which interacts with TBP?

Changes in abundance of cofactors might affect DNA accessibility, by presenting more or less DNA binding sites for a TF, or abundance changes might affect the kinetic rate constants. As stated above, we now extract the apparent number of chromatin binding sites using our measurements. The increase in apparent number of chromatin binding sites we observe for mEos2-TBP might in part be due to changes in TAF, TFIID or Pol II abundance. Discriminating all possible effects that might enter a change in accessible DNA binding sites is out of the scope of the present manuscript. For the dissociation rate constant, our data of the proportion of stable bound molecules do not indicate a change in dissociation rate constant during early development.

Of importance is that such biological factors do not alter the influence of a decreasing nucleus on the chromatin-bound fraction, this is still there.

Again the the model put forward by the authors, that it is due to a shrinking nucleus, is interesting and plausible. But it is one of many plausible models and the authors do no experiments to validate that their model is the right one.

Based on our analysis of the relation between bound molecules and all detected molecules in every stage (Figure 4c and d and Supplementary Figure S6), we assume that the law of mass action is valid in every stage. Consequently, we take the effect of shrinking nuclear volume on the chromatin-bound fraction of a TF as given. A causative relation between nuclear size and bound fraction is not just one possible model amongst many others to explain our data. This relation rather needs to be accounted for in any mechanism dealing with binding of transcription factors and adds to other possible effects, e.g. changes in DNA accessibility.

We feel that we did not well explain this in the previous manuscript. We now added a figure (Figure 1b) and several paragraphs to the introduction to describe this point much more clearly, and reflected on possible influences on the bound fraction.

Nevertheless, we tested whether we could artificially decrease nuclear size in Zebrafish embryos, analogous to the experiments by Jevtic et al., 2015 (please see Additional Figure A1). Interestingly, upon injection of Zebrafish Rtn4a in addition to mEos2-TBP, we did not observe significant changes to nuclear size, however huge changes to the chromatin-bound fraction of mEos2-TBP, along with huge changes in DNA accessibility. The effect of Rtn4a, at least in our case, thus seems to be rather in changing chromatin structure than nuclear size. We thus do not think that injection of Rtn4a, and probably other means of increasing nuclear size, will be a good experiment, since not only nuclear size but other, unrecognized changes are likely to occur. We thus refrained from further artificially changing nuclear size in the live Zebrafish embryo.

If the reviewers feel that adding these experiments, which we think demonstrate that injection of reticulon 4a to Zebrafish embryos is not a good way to change nuclear size without affecting other cellular parameters, will improve the manuscript, we will be happy to include these experiments.

As such, their evidence is poor and too speculative to support a firm conclusion.

As detailed in the points above, when clearly differentiating which parameter of the law of mass action will be affected by a certain physical or biological factor, it becomes more evident where our manuscript is conclusive and where it stays more speculative.

Our statement on the effect of nuclear size on the chromatin-bound fraction of TBP and Sox19b given in the title is valid. We rephrased it to “Decreasing nuclear volume concentrates DNA and facilitates

transcription factor–chromatin associations during Zebrafish genome activation”.

On the other hand, since measuring all possible factors that might influence DNA accessibility is out of the scope of this manuscript, we cannot say more on this than that apparent number of chromatin binding sites seems to increase for mEos2-TBP but seems to stay constant for mEos2-Sox19b. Since the apparent number of chromatin binding sites includes several possible contributions, e.g. DNA accessibility, binding competition and changes in association rate constant, we cannot disentangle it further. However, we mention the possibility that differences in this parameter between TBP and Sox19b might arise due to potential differences in how these factors bind to chromatin.

Since we did not further study how the chromatin-bound fraction translates further into transcription, we agree that the claim “*the shrinking nucleus is a major driving force and timer of ZGA in zebrafish embryos*” might be too overenthusiastic. We thus rephrased it to more emphasize our technical advances.

Authors say “To be able to draw conclusions also valid for endogenous TBP, we determined the fraction of chromatin-bound TBP molecules, a quantity independent of molecule concentration (Figure 2d and Methods).”. Again, this is not true. The total bound fraction is not independent of molecule concentration. In fact, over-expressing a protein affects the bound fraction and the residence time. For example, see the model of facilitated dissociation: <http://www.pnas.org/content/114/16/E3251> This is potentially troubling as the authors’ experiments rely on microinjection of exogenous mEos-TBP RNA, with no indication as to how much is translated and therefore they have no idea of the concentration of their exogenous construct or the amount of TBP in the embryo.

We thank the reviewer for pointing out the possible factor of a concentration-dependent dissociation rate constant that might influence the chromatin-bound fraction. We now include this factor in the paragraphs on the law of mass action in the introduction.

As detailed above, we now added a Western Blot to measure relative expression levels of mEos2-TBP and endogenous TBP and found they only differed marginally (Figure 2c).

Further, we performed additional analysis on our data to look for a hint to such a mechanism. The plots of chromatin-bound molecules to all detected molecules are linear in every stage (Figure 4c and d and Supplementary Figure S6). Facilitated dissociation would result in faster off-rates at high molecule concentration, thus the plot of bound to all detected molecules would start to level off horizontally. Such a behaviour is not compatible with our data.

And how does Figure 2d show that their conclusion is valid for endogenous TBP? How did they measure endogenous TBP? The figure legend for 2d does not mention endogenous TBP.

The reviewer is right that we cannot measure endogenous TBP.

We now added several experiments to test our mEos2-TF constructs. As detailed above, we added a Western blot analysis of mEos2-TBP showing only a marginal overexpression compared to endogenous TBP (Figure 2c). In addition, we tested whether mEos2-TBP bound to the same genes as endogenous TBP by ChIP-qPCR (Figure 2d). We note that our mEos2-TBP is closely related to a recently published CRISPR knock-in fusion protein of TBP in mammalian cells, where they were functional (Teves et al., bioRxiv 2018) and previous publications using a FP-TBP fusion construct (e.g. Chen et al., 2002). For mEos2-Sox19b, we observed preferred binding to specific genes compared to a genomic control (Figure 2e). We also added new analysis (Figure 4c and d and Supplementary Figure S6) indicating that our mEos2-TFs do not show signs of saturation.

Overall, these experiments indicate that our mEos2-TFs approximate binding of endogenous TFs. Thus, the chromatin-bound fraction of mEos2-TFs will likely approximate the chromatin-bound fraction of endogenous TFs. We adjusted our statement.

Authors state that “This result also holds for endogenous TBP, as the fraction of specifically bound molecules is independent of concentration.” We disagree. The fraction of specifically bound molecules often change with over-expression as shown in many studies, so this statement is clearly wrong as documented by many examples.

As detailed above, our experiments and analysis indicate that our mEos2-TFs approximate binding of endogenous TFs. We adjusted our statement.

Second, authors have not shown any data on endogenous TBP and they have not demonstrated that mEos2-TBP is functional. It is perfectly possible for endogenous TBP to be more functional and bound to a higher extent than mEos2-TBP. Authors should not make strong statements when they do not have any evidence to support them.

As suggested by the reviewer, we have added several experiments and data analysis. As mentioned above, we included ChIP-qPCR experiments suggesting binding of mEos2-TBP to genes of endogenous TBP binding (Figure 2d). mEos2-Sox19b showed preferred binding to specific sites compared to a genomic control (Figure 2e).

Moreover, we tested whether injection of mRNA encoding for mEos2-TBP and mEos2-Sox19b changed embryo morphology or timing. Injection of mEos2-TF did not affect the development of embryos up to 24 hpf for mEos2-TBP and up to sphere stage for mEos2-Sox19b. The developmental delay observed for embryos into which Sox19b mRNA was injected is comparable to previous observations with embryos into which Sox3, which is functionally related to Sox19b, was injected (Shih et al., 2010) (Figure 2f).

In addition, mEos2-TBP has been designed similar to previously reported FP-fusion proteins of TBP that have been shown to be functional (Chen et al., 2002, Teves et al., 2018).

These experiments indicate that our mEos2-TFs approximate binding of endogenous factors. We adjusted our statements accordingly.

Authors say that “Overall, the number of accessible DNA binding sites did not change significantly during replication (Figure 4b). This result suggests that DNA becomes compacted during replication such that sites accessible for TBP molecules approximately halve.” It is not clear how the authors measured accessible sites for TBP. It seems that it is solely based on the single molecule imaging and fraction bound of TBP. This is not an appropriate measure for DNA accessibility. Moreover, the author say they did the second set of measurements “post-replication”, so it should not matter what happened during replication since the measurement was taken after replication. Presumably they compare G1 to G2. It would be extremely surprising if the genome was 2-fold more compacted in G2 than in G1 and the authors should provide independent and compelling evidence if they want to make such a striking claim.

We agree that we did not describe this analysis very well. Since this analysis does not add any value to the current manuscript, we now omit it.

TBP residence time

The authors use time-lapse imaging (Gebhardt Nature Methods 2013) to measure the residence time. They find a residence time of ~7 seconds for TBP. This is surprisingly short given what has been previously published. Authors cite a yeast paper that reported a similar value, but other studies report 1-50 minutes (<https://www.nature.com/articles/srep39631>). In particular, Zebrafish is probably more similar to mammalian cells than yeast cells and previous FRAP on mammalian TBP found residence times on the order of minutes. It is concern that due to drift and photo-bleaching, the authors cannot measure stable (minute) binding events.

We agree that our measurements of residence time on chromatin for TBP differ to some publications, but not all. The discrepancy likely arises due to differences in TBP binding between species, or due to fast cell cycles in early Zebrafish development. Fast TBP cycling of ~7s that we observe in Zebrafish embryos is compatible with measurements of RNA Polymerase II dynamics in live cells that suggest fast transcription initiation times of ~8 s{Cho:2016wj}.

Our time-lapse approach successfully separates photobleaching from unbinding, even if photobleaching is fast (Gebhardt et al., 2013). We are able to observe residence times up to several minutes using this single molecule tracking approach (Agarwal et al., 2017).

Drift indeed occurs for cells in the Zebrafish embryos. We were aware of this and carefully inspected our data for possible drift, before including movies into analysis. Our measurements do not provide a hint for minute-long binding of molecules in interphase cells. However, we cannot completely exclude a potential contribution of unrecognized drift and out-of-focus movement of bound molecules during dark times to the observed residence time, which might lead to a potential underestimation of our residence time.

We added a corresponding discussion mentioning a possible effect of unrecognized drift or out-of-focus movement on our residence time measurements to the results section.

Under the imaging conditions used by the authors, what is the photobleaching rate or half-life? The authors should state this number in the main text.

The photobleaching rate constant was 7.3 ± 0.2 /s for mEos2-TBP, and 5.8 ± 0.2 /s for mEos2-Sox19b. We now give these numbers in the text.

Also, the authors should either be explicit about the uncertainty of their residence time measurement or provide evidence for the same result with an orthogonal technique like FRAP.

The reviewer is right in that it is important to name possible limitations to the measurements performed. As suggested, we now discuss possible effects of unrecognized drift or out-of-focus movement on our residence time measurements.

Figures showing raw data

The authors show almost no raw data in the main figures or the supplement. Figure 2e is a good example. It nicely shows all the data in one place, but it is impossible to read the figure and they do not show legible figures in the supplement. They should show the same data in a log-log or spread out manner in the supplement. It is not possible to see what the frequency of ~40 s binding events were.

As suggested we now show more raw data in the main text and Supplementary Information. In particular, we added raw data to the time-lapse approach in Figure 3b, the interlaced time-lapse approach in Figure 4c and d (and Supplementary Figure S6). We plot the histograms of visible fluorescence for mEos2-TBP and mEos2-Sox19b now with log-log scaling. We only measured one event of ~40 s binding of TBP in the 8s dark time condition, corresponding to 4 consecutive images, although we could frequently detected molecules for 5 consecutive images before bleaching at shorter time lapse conditions.

Also given the relatively high density of molecules in the movies and the frequent case of molecules coming into focus from out of focus, it is not clear the authors can properly do the tracking when there are long dark times.

We understand the concern of the reviewer. We have tested our detection algorithm using simulated data of similar SNR and density including noise (Chenouard et al., Nat Methods 2014). From these tests we estimate that our detection error of molecules is below 10%. We now state this in the

Methods section.

What do the numbers above the blue points mean in 2e? Is that the dark time? Too little detail is given.

We apologize if the description of figures was not always complete. These numbers indicate the total time-lapse time. As suggested, we now improved the description of our figures.

Technical concerns

· Authors inject mEos2-TBP RNA molecules. What is their evidence that the mEos2-TBP fusion protein is fully functional? This is a major concern, because the fusion protein is expressed in a background where the wild-type protein is also expressed.

As detailed above, we now included Western blot analysis of mEos2-TBP expression compared to endogenous TBP expression, showing only marginal overexpression. We also added ChIP-qPCR experiments suggesting that mEos2-TBP binds to genes of endogenous TBP binding (Figure 2d). In addition, our mEos2-TBP fusion protein is similar in design to a GFP-TBP published earlier (Chen et al., 2002) and a functional CRISPR knock-in of Halo-TBP in mammalian cells (Teves et al., 2018).

We also added ChIP-qPCR experiments for meos2-Sox19b demonstrating preferred binding to specific genes compared to a genomic control sequence (Figure 2d). A comparison to endogenous Sox19b binding could unfortunately not be performed due to lack of a good Sox19b antibody. This also prevented a test for overexpression, however, we used only half the injected amount of mRNA compared to mEos2-TBP, and did not observe a sign of saturation in the plots of bound molecules vs all detected molecules in each stage (Figure 4d and Supplementary Figure S6).

Moreover, we tested whether injection of mRNA encoding for mEos2-TBP and mEos2-Sox19b changed embryo morphology or timing. Injection of mEos2-TF did not affect the development of embryos up to 24 hpf for mEos2-TBP and up to sphere stage for mEos2-Sox19b. The developmental delay observed for embryos into which Sox19b mRNA was injected is comparable to previous observations with embryos into which Sox3, which is functionally related to Sox19b, was injected (Shih et al., 2010) (Figure 2f).

· Authors say mEos2-TBP concentration increases by about ~25 fold from 64-cell stage to oblong stage. How do the author know that translation of their injected mRNA is physiologically regulated. The simplest model is that mRNA is constantly translated and the longer you wait, the more protein you get instead of this being a case of physiological regulation.

We agree with the reviewer that our measurements of mEos2-TBP concentration do not reflect the concentration of endogenous TBP. We therefore designed our study such that these differences do not affect our analysis or conclusions. We now state this fact more clearly and refrain from reporting concentrations in the main text, to prevent possible confusions.

Nevertheless, our methodology allows accurate measurements of the concentration of fluorescent fusion proteins. This will become important if for example CRISPR knock-ins become more generally available. We thus still report the concentration measurements in the supplements.

· There are a lot of concerns about the concentration measurements. And the authors do not explain in sufficient detail how they did them. They say they change the photoactivation period depending on the mEos2 concentration – this means that they compare different conditions? Also in a 0.5s vs. a 5s photoactivation period, non-photo-activated mEos molecules can move in and out of focus of the activation laser lightsheet to different extents giving rise to complicated relationships between pulse duration and extend of photo-activation.

The reviewer raises an important point. We initially applied two different measurement protocols. In the first, the power of the activation laser was controlled and similar for each developmental stage. For concentration measurements, we only applied this protocol. In the other measurement protocol, this power was not controlled and changed between stages, which added a systematic detection noise to our measurement. We had used such a protocol for a part of the measurements of the chromatin-bound fraction of mEos2-TBP. We now restricted our analysis to the condition with controlled activation power for all measurements to avoid this systematic instrumental noise.

· *How much embryo-to-embryo variability in the amount of injected mRNA was there? How did authors control for this? How did authors ensure that mRNA was uniformly distributed to avoid high local concentration gradients? Diffusion constants of mRNAs are very low, so this could be a big problem.*

Variability of mRNA injection volume was ca. 15%. As suggested, we now give more details on the injection protocol in the Methods section.

As detailed above, the chromatin-bound fraction is independent of concentration. Thus, embryo-to-embryo or cell-to-cell variability or inhomogeneous distribution of mRNA do not affect this measurement. In addition, we performed our measurements in several embryos, and the values of the chromatin-bound fraction are reported as average \pm sem.

· *In single-molecule imaging it is critical to correct for photobleaching, which does not happen during the dark periods, and which is obviously important to correct for. But living embryos move a lot and molecules can also move out-of-focus and disappear further complicating such corrections. Drift, unlike photobleaching, should depend on time and thus increase with increasing dark time. The residence time fitting model ignores this and I think this might be why they claim to calculate such a short TBP residence time. Authors should correct for drift carefully.*

Our time-lapse approach is very well suited to separate photo-bleaching from dissociation.

Regarding drift, we agree that this might lead to artificial cut offs of bound molecules. We were aware of this and thus have inspected our data carefully for drift before analysis. We did not include measurements with drift into our analysis. Our measurements do not provide a hint for minute-long binding of molecules in interphase cells. However, we cannot completely exclude a potential contribution of unrecognized drift and out-of-focus movement of bound molecules during dark times to the observed residence time, which might lead to a potential underestimation of our residence time.

We now include a corresponding discussion in the text.

· *Why are the units for the number of accessible DNA sites (3b) in AU? It should be number per genome (e.g. 7000 per genome).*

There is currently no possibility to calibrate the value for the apparent number of chromatin binding sites. We thus give this value as relative values normalized to the 64-cell stage.

Additional Figure

Additional Figure A1. The chromatin-bound fraction of mEos2-TBP in absence and presence of Rtn4a. **(a)** Chromatin-bound fraction of mEos2-TBP in absence (blue) and presence (red) of Rtn4a during development (mean \pm s.e.m.). Lines represent the chromatin-bound fraction as calculated from the law of mass action using data from **(b)** and **(c)** and extracting the apparent number of chromatin binding sites as free parameter. Shades represent error intervals propagated from the errors in stable bound proportion and nuclear volume. The panel includes in total 135945 molecules from 6 embryos for mEos2-TBP in absence and 171498 molecules from 3 embryos for mEos2-TBP in presence of Rtn4a. **(b)** Stable bound proportion of chromatin-bound mEos2-TBP in absence (blue) and presence (red) of Rtn4a during development (mean \pm s.e.m.). The panel includes in total 6850 molecules from 6 embryos for mEos2-TBP in absence and 12462 molecules from 3 embryos for mEos2-TBP in presence of Rtn4a. The dashed line is given as guide to the eye. **(c)** Nuclear volume of embryos injected with mRNA for mEos2-TBP in absence (blue, $n=4$) and presence (red, $n=3$) of Rtn4a during development (mean \pm s.e.m.). The dashed line is given as guide to the eye. **(d)** Apparent number of chromatin binding sites of mEos2-TBP in absence (blue) and presence (red) of Rtn4a as calculated from the law of mass action using data from **(a)**, **(b)** and **(c)**. Shades represent error intervals propagated from the errors in chromatin-bound fraction, stable bound proportion and nuclear volume.

Reviewers' Comments:

Reviewer #1:

Remarks to the Author:

This is a massively improved manuscript, which addressed most of the major concerns that were raised on the first version. Several of the criticised experiments were removed and the narrative streamlined. The overinterpreted claims have mostly been removed and as a result, the ms is much tighter and presents a clearer and better substantiated message. There are new additions, such as imaging and analysis of a fluorescently labelled Sox19 transcription factor, which together with the TBP strengthens their model on the effect of nuclear size on chromatin bound fraction of transcription factors during cleavage stage of development.

The most important concern regarding the equivalence of overexpressed TBP to that of the endogenous remains problematic. There are 3 additional experiments, which are supposed to address this and other reviewers' concerns but neither of them is satisfactory

1. Western blots. The authors argue for marginal difference between endogenous and overexpressed protein levels. Firstly, given the dynamic nature of endogenous TBP expression, a single stage does not address this matter sufficiently. All stages used in the experiments ought to be compared for satisfactory conclusion on comparability of levels. Secondly, what the authors consider marginal, looks to this reviewer as 2-3x difference.

2. Overexpression phenotypes. These are promising, however only make sense with number of embryos analysed and % of embryos showing the depicted phenotypes.

3. ChIP. The CHIP experiments for TBP are poorly executed and lack appropriate controls. Two different antibodies (HA and TBP) with two different epitopes cannot be compared directly, they need to be tested by comparison of promoter bound fraction to that bound to negative sites in the genome (for example intragenic and intergenic random sites).

Given that the role of nuclear size in TF binding frequency is based on overexpressed proteins with a number of identified unknown variables and without interference experiments this study remains limited in its scope to a model, which is while elegant, not yet proven to be valid on endogenous proteins. Therefore, the authors need to explicitly state in the abstract that their work is carried out with fluorescently labelled TBP and Sox19b proteins and the title should be more carefully phrased to reflect that their conclusion is a feasible model rather than proof of causative interaction. If these remaining concerns are addressed the manuscript will be suitable for publication.

Minor points:

The word zebrafish is to be typed with small caps in text and title

pCS2+ plasmid is cited consistently, however this vector is probably the commonly used expression vector pCS2+.

There are a number of typos in the ms that need to be corrected.

The TBP sequence inserted into pCS2+ remains unknown, only a protein coding peptide sequence is referred to without DNA fragment info.

Reviewer #2:

Remarks to the Author:

The authors have addressed my concerns.

Reviewer #3:

Remarks to the Author:

Review of "Decreasing nuclear volume concentrates DNA and facilitates transcription factor-chromatin associations during Zebrafish genome activation" by Reisser et al.

Main summary

The revised manuscript from Reisser et al. report an elegant series of imaging techniques that allowed the authors to visualize individual molecules in Zebrafish embryos. The main findings can be summarized as follows:

1. A newly developed Reflected Light Sheet Microscopy allowed visualization individual TF molecules in live embryos.
2. TBP and Sox19b showed different dissociation rates using RLSM.
3. The fraction of bound TFs increases over developmental stages in the embryo.

The authors have made significant changes from the previous manuscript. They have added new experiments and analyses, expanded their methods and description of experiments in the main text and figures, and toned down statements that may not be directly supported by their data. They also removed several experiments that were problematic or over-interpreted in the first manuscript. However, several concerns that we presented in the first review remain unaddressed.

Major concerns:

1. We were originally concerned that the authors' argument in the first manuscript that nuclear volume is the primary driver of TF binding (paraphrasing) is more likely only one of many potential contributors to the observed increase in TF binding during development (others include chromatin organization, increased transcription, increased TF expression, expression of developmental cofactors, etc). We appreciate that the authors have toned down their causative statements in the revised manuscript, especially since they have not performed perturbation assays to test for causation as previously suggested by multiple reviewers. However, the authors argue that the "In a framework assuming equilibrium, TF-chromatin binding is governed by the law of mass action" (p. 2) and use this argument as the basis for many of their analyses and interpretation of their data. First, a living cellular environment largely does not exist in equilibrium. The fact that many processes require energy input (in the form of ATP) would highly suggest that cellular environment, and hence nuclear environment, is not in equilibrium. Therefore, the law of mass action may not always apply. We also note that the third paragraph of the Introduction where the authors further expand on the law of mass action has no references to published studies indicating applicability of the law of mass action in biological systems. Second, the increasing evidence for phase-separated domains or protein-protein interaction hubs that increase local concentration would suggest that global nuclear concentration of TFs (as directly related to nuclear volume) is not linearly related to TF binding. For example, the morphogen Bicoid is seen to form clusters of binding even in the posterior of the drosophila embryo where global intra-nuclear levels have decreased exponentially to vanishingly low levels (<http://genesdev.cshlp.org/content/31/17/1784.abstract>). We also note that Drosophila embryos represent another developmental system where nuclear volume decreases over each cell cycle event and yet, given the studies about local high concentration TF domains that are dependent on TF-TF interactions (i.e. Bicoid-Zelda) nuclear volume would not seem to be the primary driver of TF binding.

2. A second major concern that we had in the original manuscript is the authors' interpretations of their experiments using exogenously expressed and tagged TBP as reflective of endogenous TBP, which is not possible to claim without extensive testing. In particular, it is troubling to make such conclusions without providing functional analyses on the exogenous-tagged TBP. We were also concerned with the authors' claim that the observed fraction bound for overexpressed TBP is reflective of the endogenous TBP because fraction bound is independent of concentration. We appreciate that the authors have now added Western blot analysis of their overexpressed construct as well as developmental assays to show that at least overexpression of TBP has no major effect on developmental timing. They cite a Biorxiv preprint on endogenously tagged TBP in mouse embryonic stem cells being functional as support for their argument that their exogenous TBP is functional, but this same preprint shows that overexpressed TBP in mouse ES cells does not

behave the same way as endogenous TBP, which is precisely the opposite of what the authors are claiming. Also problematic is the authors' persistent claim that the fraction of bound molecules is independent of concentration. The ON-rate should be $\text{rate}_{\text{on}} = k_{\text{on}} [\text{TF}][\text{DNA}]$. This means that the fraction bound, which is related to first-order rate constants ($k_{\text{on}}/(k_{\text{on}} + k_{\text{off}})$) is directly related to the concentration of free molecules. In contrast, the rate of unbinding (dissociation constant) $\text{rate}_{\text{off}} = k_{\text{off}} [\text{TF-DNA}]$, which means dissociation is not dependent on global TF molecule concentration. Regardless of the mathematical derivation shown in Supplemental Info (ex. Equation 28), the fact that the fraction bound is directly related to concentration of TFs cannot be ignored. One example in the literature of overexpressing proteins affecting fraction bound has already been presented in the case of CTCF (<https://elifesciences.org/articles/25776>). This directly contradicts the claim the authors make. For other examples of recent studies demonstrating that over-expressed proteins do not behave like endogenous proteins please see this recent review: [https://www.cell.com/biophysj/fulltext/S0006-3495\(18\)30619-2](https://www.cell.com/biophysj/fulltext/S0006-3495(18)30619-2) Furthermore, the authors' logic that the fraction bound is independent of concentration seems in direct contradiction to their favored hypothesis that decreasing nuclear volume promotes increased TF binding. Indeed, as the authors write "Increasing TF levels will increase the absolute number of chromatin-bound TF molecules in non-saturating conditions" (p. 2). Another point is facilitated dissociation: John Marko's lab has shown that several DNA-binding proteins exhibit concentration-dependent k_{OFF} rates. This suggests that both the k_{ON} and k_{OFF} rate could be concentration dependent. In summary, there are many reasons to suspect that over-expressed TBP will not behave like endogenous TBP, and perhaps the even for Sox19b.

Minor concerns:

1. When the authors write "Within all stages, the relation between chromatin-bound and all mEos2-TF molecules was linear, as expected from the law of mass action. This indicates that saturation effects or facilitated dissociation do not occur" (p. 6), we agree that saturation is likely not a factor, but the linear increase in fraction bound says nothing about occurrence or non-occurrence of facilitated dissociation
2. The authors state that "60 pg of mRNA only marginally increased mEos2-TBP expression over endogenous levels" (p. 4). However, the Western blot shows that the levels of tagged TBP is at least 3-4x the endogenous level (300-400%). Quantifying the Western blot bands would provide more reliable numbers. Also, where is the Western analysis of mEos2-Sox19b?

Reviewer #4:

None

Response to reviewers' comments:

Reviewer #1 (Remarks to the Author):

We thank the reviewer for their constructive comments. We here respond to every point. If appropriate, we state in red the text that we added to the new manuscript or refer to the corresponding section.

We have performed more controls regarding the analysis of protein expression levels during development by Western Blot, give the full statistics of phenotype determination and added new CHIP experiments on TBP binding. New data are reported in existing figures or in new supplemental figures. We also changed the title and abstract to better reflect our findings.

1. Western blots. The authors argue for marginal difference between endogenous and overexpressed protein levels. Firstly, given the dynamic nature of endogenous TBP expression, a single stage does not address this matter sufficiently. All stages used in the experiments ought to be compared for satisfactory conclusion on comparability of levels. Secondly, what the authors consider marginal, looks to this reviewer as 2-3x difference.

To address this concern we performed Western Blot analysis on the 64-cell stage up to the oblong stage. Both endogenous TBP and ectopic mEos2-TBP abundance increased from 64-cell to oblong stage. We included the corresponding gels in the revised manuscript in Supplementary figure S2. Additionally we quantified the expression levels in the 1k cell stage on three biological replicates, of which two also were technically replicated. We find that in 1k stage the mEos2-TBP expression is 3.8 ± 2.5 fold compared to the endogenous TBP level. We now omitted the phrase „only marginally”.

We added the following description to the manuscript:

60 pg of mRNA increased ectopic mEos2-TBP expression at the 1000 cell stage 3.8 ± 2.5 (mean \pm std) fold over endogenous TBP levels, as shown by Western Blot (Figure 2c, Supplementary Figure S2 and Methods). Both endogenous TBP and ectopic mEos2-TBP abundance increased from 64-cell to oblong stage.

2. Overexpression phenotypes. These are promising, however only make sense with number of embryos analysed and % of embryos showing the depicted phenotypes.

We now give the full statistic of the overexpression phenotypes in the new Supplementary figure S3. In addition, we replaced the images at 24hpf to follow the convention in fish orientation.

We extended the description of phenotypes in the text:

Injection of mEos2-TBP did not affect the development of embryos into prim-5 stage at 24 hpf (Figure 2f and Supplementary Figure S3). For mEos2-Sox19b, no developmental phenotype was visible up to sphere stage, thus suggesting normal behaviour during our measurement period up to oblong stage. Thereafter, mEos2-sox19b injected embryos displayed developmental delay but no overt phenotype at mid-somitogenesis stages. At the prim-5 stage defects in development of posterior structures became obvious.

3. CHIP. The CHIP experiments for TBP are poorly executed and lack appropriate controls.

Two different antibodies (HA and TBP) with two different epitopes cannot be compared directly, they need to be tested by comparison of promoter bound fraction to that bound to negative sites in the genome (for example intragenic and intergenic random sites).

As suggested, we have set up a new ChIP experiment including two promoter regions of genes expressed in the early embryo (hmga1a and brd2a) and an intragenic negative control. The experiment shows that 2xHA-mEos2-TBP is enriched in promoter regions compared to the control.

We changed the description of the ChIP experiment to:

For mEos2-TBP, we observed preferential binding to promoters of genes expressed in the early embryo⁴ as compared to a genomic control by ChIP-qPCR (Figure 2d and Methods). mEos2-Sox19b bound preferentially to Sox19b-specific target sites as compared to a genomic control (Figure 2e).

Given that the role of nuclear size in TF binding frequency is based on overexpressed proteins with a number of identified unknown variables and without interference experiments this study remains limited in its scope to a model, which is while elegant, not yet proven to be valid on endogenous proteins. Therefore, the authors need to explicitly state in the abstract that their work is carried out with fluorescently labelled TBP and Sox19b proteins

As suggested, we included the information of fluorescently labelled proteins in the abstract:

... to visualize two fluorescently labelled transcription factor (TF) species, mEos2-TBP and mEos2-Sox19b.

In addition we state similarity to endogenous protein binding more carefully in the main text:

Overall, these experiments indicate that chromatin-binding of mEos2-TFs approximate binding of endogenous TF within the limitations imposed by ectopic expression and an added fluorescent tag.

and the title should be more carefully phrased to reflect that their conclusion is a feasible model rather than proof of causative interaction.

We now changed the title to "Single molecule imaging correlates decreasing nuclear volume with increasing transcription factor–chromatin associations during zebrafish genome activation". This emphasizes our technical approach and the fact that we cannot proof the correlation nuclear volume – increased bound fraction, although dictated by physics, with a bijective perturbation experiment in the in vivo setting of the embryo. A true bijective experiment, that exclusively changes nuclear volume without any other changes for example in chromatin organization or cytoplasmic volume is not possible with current technology.

If these remaining concerns are addressed the manuscript will be suitable for publication.

Minor points:

The word zebrafish is to be typed with small caps in text and title

We corrected our manuscript for this typo.

pSC2+ plasmid is cited consistently, however this vector is probably the commonly used expression vector pCS2+.

We apologize for the mix-up and corrected our text accordingly.

There are a number of typos in the ms that need to be corrected.

We carefully checked the manuscript for typos.

The TBP sequence inserted into pCS2+ remains unknown, only a protein coding peptide sequence is referred to without DNA fragment info.

We now added the complete nucleotide sequence of the TBP we used to the supplementary information as Supplementary Table 4.

Reviewer #2 (Remarks to the Author):

The authors have addressed my concerns.

We thank the reviewer for appreciating our work.

Reviewer #3 (Remarks to the Author):

We thank the reviewer for their comments. We here respond to every point. For clarity, we introduced sub sectioning i), ii), iii),.... If appropriate, we state in red the text that we added to the new manuscript or refer to the corresponding section.

We have performed more controls regarding the analysis of protein expression levels during development by Western Blot, give the full statistics of phenotype determination and added new ChIP experiments on TBP binding. New data are reported in existing figures or in new supplemental figures. We also changed the title and abstract to better reflect our findings.

Major concerns:

1. i) We were originally concerned that the authors' argument in the first manuscript that nuclear volume is the primary driver of TF binding (paraphrasing) is more likely only one of many potential contributors to the observed increase in TF binding during development (others include chromatin organization, increased transcription, increased TF expression, expression of developmental cofactors, etc).

We appreciate that we now could clarify our view of a coexistence of influencing factors, where the decreasing nuclear volume acts in addition to a manifold of biologically imposed effects.

ii) We appreciate that the authors have toned down their causative statements in the revised manuscript, especially since they have not performed perturbation assays to test for causation as previously suggested by multiple reviewers.

We agree that the influence of nuclear volume, while inevitable in any comprehensive physical description, could still be demonstrated by a perturbation experiment. As shown in our last response, we tried to repeat previous efforts in this direction. However, we came to realize that a true bijective perturbation experiment in the in vivo setting of the embryo, that exclusively changes nuclear volume without changes of any other kind, for example in chromatin organization or cytoplasmic volume, is not possible with current technology. We therefore rephrased the title to “Single molecule imaging correlates decreasing nuclear volume with increasing transcription factor–chromatin associations during zebrafish genome activation”.

iii) However, the authors argue that the “In a framework assuming equilibrium, TF-chromatin binding is governed by the law of mass action” (p. 2) and use this argument as the basis for many of their analyses and interpretation of their data. First, a living cellular environment largely does not exist in equilibrium. The fact that many processes require energy input (in the form of ATP) would highly suggest that cellular environment, and hence nuclear environment, is not in equilibrium. Therefore, the law of mass action may not always apply. We also note that the third paragraph of the Introduction where the authors further expand on the law of mass action has no references to published studies indicating applicability of the law of mass action in biological systems.

We agree that living systems overall operate out of equilibrium and many biological systems could be found in which the law of mass action is not directly valid. We already had mentioned this in our manuscript (“...non-equilibrium processes typically observed in biological systems...” in Introduction, paragraph three). In particular, while often assumed (e.g. Berg et al., *Biochemistry* 20, 6929–6948 (1981) or Li et al., *Nat. Phys.* 5, 294–297 (2009)), it has not been demonstrated that TF-chromatin interactions in the nucleus indeed are at equilibrium and follow the law of mass action.

However, we disagree that we naively use this as basis of data analysis and interpretation. We already in the last version of our manuscript have presented a detailed analysis of the measured relation between TF number and TF binding (Fig 4c and d and Supplementary Fig S8). This analysis did not reveal any sign that the law of mass action would not be valid within every stage. Thus, for the sake of scientific parsimony, we conclude that TBP and Sox19b interactions with chromatin in every stage occur at equilibrium and follow the law of mass action.

Our approach to first in general discuss consequences of an assumption in the introduction and then later test whether the assumption was a good one, as we do for mEos2-TBP and mEos2-Sox19b, is a valid approach.

We stress the experimental validation of our initial assumption now more clearly in the text:

We then first tested whether our initial assumption, that TFs bind to chromatin following the law of mass action was indeed true for mEos2-TBP and mEos2-Sox19b or whether effects of saturation or facilitated dissociation would be visible.

In addition, we restrict the assumption of equilibrium now to TF-chromatin interactions:

In a framework assuming equilibrium of TF-chromatin interactions, TF-chromatin binding is governed by the law of mass action.

iv) Second, the increasing evidence for phase-separated domains or protein-protein interaction hubs that increase local concentration would suggest that global nuclear concentration of TFs (as directly related to nuclear volume) is not linearly related to TF binding. For example, the morphogen Bicoid is seen to form clusters of binding even in the posterior of the drosophila embryo where global intra-nuclear levels have decreased exponentially to vanishingly low levels (<http://genesdev.cshlp.org/content/31/17/1784.abstract>). We also note that Drosophila embryos represent another developmental system where nuclear volume decreases over each cell cycle event and yet, given the studies about local high concentration TF domains that are dependent on TF-TF interactions (i.e. Bicoid-Zelda) nuclear volume would not seem to be the primary driver of TF binding.

While such domains or hubs might exist for some proteins, for example Bicoid as mentioned by the reviewer, we do not observe any sign of hub formation of TBP and Sox19b. Together with the measured linear behaviour in the relation between TF number and bound TF, as detailed above, we take the simplest model explaining our data, which is the law of mass action.

We agree with the reviewer that in the special example of Bicoid binding in Drosophila, interesting biological boundary conditions to the law of mass action are in place, such as the local concentration of six Bicoid binding sites in the promoter of the hunchback gene or Bicoid-Zelda interactions. Thus, for this system, a detailed model would have to take these biological boundary conditions into account, together with the decreasing nuclear volume. However, since the authors of the cited paper did not investigate changes of Bicoid binding between different developmental stages, a speculation about the relative impact of nuclear volume compared to other biological influences in this system is futile.

We now include this example in the discussion:

We note that for each TF specific biological constraints, such as formation of local high density hubs⁶⁸ need to be considered in addition to the effect of nuclear size.

2. i) A second major concern that we had in the original manuscript is the authors' interpretations of their experiments using exogenously expressed and tagged TBP as reflective of endogenous TBP, which is not possible to claim without extensive testing. In particular, it is troubling to make such conclusions without providing functional analyses on the exogenous-tagged TBP. We were also concerned with the authors' claim that the observed fraction bound for overexpressed TBP is reflective of the endogenous TBP because fraction bound is independent of concentration. We appreciate that the authors have now added Western blot analysis of their overexpressed construct as well as developmental assays to show that at least overexpression of TBP has no major effect on developmental timing.

We are happy to see that the reviewer appreciates the Western Blot and phenotyping controls we have performed for TBP.

We even further extended the Western Blot controls, to show how endogenous and ectopically expressed TBP behave between the 64-cell stage and the oblong stage. Both endogenous TBP and ectopic mEos2-TBP abundance increased from 64-cell to oblong stage. We included the corresponding gels in Supplementary figure S2. Additionally we quantified the expression levels in the 1k cell stage on three biological replicates, of which

two also were technically replicated. We find that in 1k stage the mEos2-TBP expression is 3.8 ± 2.5 fold compared to the endogenous TBP level. We now omitted the phrase „only marginally“.

We added the following description to the manuscript:

60 pg of mRNA increased ectopic mEos2-TBP expression at the 1000 cell stage 3.8 ± 2.5 (mean \pm std) fold over endogenous TBP levels, as shown by Western Blot (Figure 2c, Supplementary Figure S2 and Methods). Both endogenous TBP and ectopic mEos2-TBP abundance increased from 64-cell to oblong stage.

ii) They cite a Biorxiv preprint on endogenously tagged TBP in mouse embryonic stem cells being functional as support for their argument that their exogenous TBP is functional, but this same preprint shows that overexpressed TBP in mouse ES cells does not behave the same way as endogenous TBP, which is precisely the opposite of what the authors are claiming.

The cited paper demonstrates nicely that tagged TBP is functional in a sense that knocked-in tagged TBP fulfils the requirements of the cell to survive. Thus, we can conclude that DNA binding and incorporation into the transcription complex are biologically functional for tagged TBP.

We now emphasize this more clearly in the text:

It has been shown that a Halo-tag knocked in to the N-terminus does not compromise TBP and Sox2 function^{39,40}.

We further control for a proper behaviour of tagged TBP by presenting new ChIP experiments including two promoter regions of genes expressed in the early embryo (hmga1a and brd2a) and an intragenic negative control. The experiment shows that 2xHA-mEos2-TBP is enriched in promoter regions compared to the control.

We changed the description of the ChIP experiment in the manuscript to:

For mEos2-TBP, we observed preferential binding to promoters of genes expressed in the early embryo⁴ as compared to a genomic control by ChIP-qPCR (Figure 2d and Methods). mEos2-Sox19b bound preferentially to Sox19b-specific target sites as compared to a genomic control (Figure 2e).

Regarding the differences in behaviour between overexpressed Halo-TBP and endogenous Halo-TBP in Teves et al., eLife 2018, these differences occur in fixed cells in mitosis, but not in live cells in mitosis (their Fig 1F). Since we measure the behaviour of ectopically expressed mEos2-TBP in interphase in live cells, we do not see an argument against the proper functionality of our mEos2-TBP here.

iii) Also problematic is the authors' persistent claim that the fraction of bound molecules is independent of concentration. The ON-rate should be $\text{rate}_{\text{on}} = k_{\text{on}} [\text{TF}][\text{DNA}]$. This means that the fraction bound, which is related to first-order rate constants ($k_{\text{on}}/(k_{\text{on}} + k_{\text{off}})$) is directly related to the concentration of free molecules. In contrast, the rate of unbinding (dissociation constant) $\text{rate}_{\text{off}} = k_{\text{off}} [\text{TF-DNA}]$, which means dissociation is not dependent on global TF molecule concentration.

We disagree. The formula given by the reviewer for the bound fraction is not correct. To see this, let's have a closer look at the units of the parameters, as defined by the reviewer:

- The ON-rate (rate on above) is the rate at which TF-DNA associations change the concentration of [TF-DNA] complexes. It is expressed in units of M/s
- The kinetic on-rate k_{on} gives the frequency of association events to a certain binding site given the concentration of the respective binding partner. It is therefore given in units of $1/(Ms)$
- The rate off above is the rate at which TF-DNA dissociations change the concentration of [TF-DNA] complexes. It is expressed in units of M/s
- The kinetic off-rate k_{off} gives the frequency of dissociation events from a certain binding site. It is therefore given in units of $1/s$

Given these definitions by the reviewer above, it is unclear what the entity $(k_{on}/(k_{on} + k_{off}))$ could possibly mean. It is however not the fraction bound (as the reviewer claims) as the units of k_{on} and k_{off} do not even match.

Instead, the true bound fraction is given by $(k_{on}[DNA]/(k_{off}+k_{on}[DNA]))$ (see our extended derivation in the supporting info), which is independent of TF concentration.

In addition, the reviewer uses a definition of dissociation constant (rate of unbinding), which is not consistent with the definition given in textbooks. There, dissociation constant K_d is given as $K_d = k_{off}/k_{on}$. In contrast, the rate of unbinding is also termed dissociation rate constant. Thus, care has to be taken not to confuse nomenclature. We consistently use the common definitions given in textbooks.

To further clarify the origin of our formulae, we now extended our derivations in the Supplementary Information (4.2.1, 4.2.2 and 4.2.3). Our derivation reveals that the bound fraction is indeed independent of concentration in the absence of saturation.

iv) Regardless of the mathematical derivation shown in Supplemental Info (ex. Equation 28), the fact that the fraction bound is directly related to concentration of TFs cannot be ignored.

We are surprised by this statement. Our derivation and results of the model are mathematically sound. If the reviewer disagrees with a certain point in an equation, please point this out explicitly. As we detailed above, the expression for the bound fraction given by the reviewer is not correct and thus their statement given here is wrong.

We now have further extended our derivation. The results stay as in the original manuscript, the bound fraction is independent of concentration.

v) One example in the literature of overexpressing proteins affecting fraction bound has already been presented in the case of CTCF (<https://elifesciences.org/articles/25776>). This directly contradicts the claim the authors make. For other examples of recent studies demonstrating that over-expressed proteins do not behave like endogenous proteins please see this recent review: [https://www.cell.com/biophysj/fulltext/S0006-3495\(18\)30619-2](https://www.cell.com/biophysj/fulltext/S0006-3495(18)30619-2)

We agree that CTCF can have a problem with high overexpression, as we see in our own study of Halo-CTCF binding (Agarwal et al., 2017) that cells show a clear phenotype if CTCF overexpression is high (cells tend to stall in S-Phase).

In the case of Telomerase (Zhong et al., Cell 2012, referenced to within the review cited by the referee above and citations therein), overexpression led to saturation of binding sites.

We thus carefully tested whether mEos2-TBP and mEos2-Sox19b overexpression caused a phenotype, but observed normal development up to 24hpf for mEos2-TBP and up to sphere stage for mEos2-Sox19b (Fig 1f). We now additionally give the full statistics of this experiment in Supplementary Fig S3. Moreover, we do not see a sign of binding saturation for both ectopically expressed proteins (Fig 4c and d and Supplementary Fig S8).

In the example of RNA Pol II (Cho et al., Sci. Rep. 2016, referenced to within the review cited by the referee above), the authors claim that the dynamics of endogenously tagged Pol II did not deviate from the one of exogenously overexpressed Pol II, demonstrating that overexpression is not always a problem.

Given the examples above that partly report equal dynamics between exogenously and endogenously expressed proteins, our controls are sufficient to support that chromatin-binding of both mEos2-TFs approximate binding of endogenous TF within the limitations imposed by an added fluorescent tag.

We now mention this limitation in the main text:

Overall, these experiments indicate that chromatin-binding of mEos2-TF approximates binding of endogenous TF within the limitations imposed by ectopic expression and an added fluorescent tag.

vi) Furthermore, the authors' logic that the fraction bound is independent of concentration seems in direct contradiction to their favored hypothesis that decreasing nuclear volume promotes increased TF binding. Indeed, as the authors write "Increasing TF levels will increase the absolute number of chromatin-bound TF molecules in non-saturating conditions" (p. 2).

We disagree that there is a contradiction. This comment probably arises due to a misunderstanding of the nomenclature, which already became apparent in the comment 2iii) on k_{on} and k_{off} above. The bound fraction (or percentage of bound molecules among all present molecules) is a unit-less quantity that takes a value between 0 and 1 (or 0% and 100%). This range of values is irrespective of the absolute number of molecules present. Instead, the bound fraction is a measure of how efficient molecules bind. The absolute number of bound molecules is then calculated from the absolute number of present molecules times the bound fraction.

For example: For 10 molecules present and a bound fraction of 0.1, 9 are free and 1 is bound. If absolute levels are increased, say to 100 molecules, with the same bound fraction of 0.1, already 10 molecules are bound, so the absolute number of bound molecules increased because the number of present molecules increased.

In our case, we observe that the bound fraction increases during development. Thus, even if the absolute number of molecules stayed the same, they bind more efficiently and thus the absolute number of bound molecules increases. For example, if again 100 molecules were present, but the bound fraction increased to 0.4, already 40 molecules out of the 100 would bind.

We hope that the reviewer now sees that this point of criticism is unsubstantiated.

We included the numerical example above in the discussion:

As numerical example, if initially 10 molecules were present in a nucleus at a bound fraction of 0.1, 1 molecule would be bound. If the level of the TF increased during development, for example to 100 molecules, already 10 molecules would be bound.

...

In our example above, if during development also the bound fraction increased, say to 0.4, 40 molecules instead of just 10 would be bound.

vii) Another point is facilitated dissociation: John Marko's lab has shown that several DNA-binding proteins exhibit concentration-dependent k_{OFF} rates. This suggests that both the k_{ON} and k_{OFF} rate could be concentration dependent.

We already had discussed this point in the last round of revision. Due to the suggestion of this reviewer, we had included the point of facilitated dissociation in our manuscript.

As can be seen from our discussion of bound fraction above (bound fraction = $(k_{\text{ON}}[\text{DNA}]) / (k_{\text{OFF}} + k_{\text{ON}}[\text{DNA}])$), an increase of k_{OFF} with concentration, the hallmark of facilitated dissociation (in the paper of John Marko's lab, k_{ON} is measured to be constant), would lead to a decrease of bound fraction. Transferred to our plots of bound molecules versus all detected molecules for every stage in Fig 4 and Supplementary Fig S8, a mechanism of facilitated dissociation would lead to a nonlinear behaviour (a concave curve), which we do not observe. Thus, as discussed above, the simplest model explaining our data is one where facilitated dissociation does not play a role.

We now changed our conclusion on facilitated dissociation to:

This indicates that saturation effects and facilitated dissociation are not observable, since both effects would yield a concave curve.

Minor concerns:

1. When the authors write "Within all stages, the relation between chromatin-bound and all mEos2-TF molecules was linear, as expected from the law of mass action. This indicates that saturation effects or facilitated dissociation do not occur" (p. 6), we agree that saturation is likely not a factor, but the linear increase in fraction bound says nothing about occurrence or non-occurrence of facilitated dissociation

This comment is related to point 2vii) of this reviewer. Please see our explanation there.

2. The authors state that "60 pg of mRNA only marginally increased mEos2-TBP expression over endogenous levels" (p. 4). However, the Western blot shows that the levels of tagged TBP is at least 3-4x the endogenous level (300-400%). Quantifying the Western blot bands would provide more reliable numbers.

This comment is related to point 2i) of this reviewer.

We even further extended the Western Blot controls, to show how endogenous and ectopically expressed TBP behave between the 64-cell stage and the oblong stage. Both endogenous TBP and ectopic mEos2-TBP abundance increased from 64-cell to oblong stage. We included the corresponding gels in Supplementary figure S2. Additionally we quantified the expression levels in the 1k cell stage on three biological replicates, of which two also were technically replicated. We find that in 1k stage the mEos2-TBP expression is 3.8 ± 2.5 fold compared to the endogenous TBP level. We now omitted the phrase „only marginally“.

We added the following description to the manuscript:

60 pg of mRNA increased ectopic mEos2-TBP expression at the 1000 cell stage 3.8 ± 2.5 (mean \pm std) fold over endogenous TBP levels, as shown by Western Blot (Figure 2c, Supplementary Figure S2 and Methods). Both endogenous TBP and ectopic mEos2-TBP abundance increased from 64-cell to oblong stage.

Also, where is the Western analysis of mEos2-Sox19b?

We already discussed this point in the first round of review.

There is no antibody that can distinguish between Sox19b and other members of the Sox family. Thus, exclusive quantification of Sox19b is not possible and the approach of Western Blot is useless for Sox19b.

However, as with mEos2-TBP, the fact that (i) mEos2-Sox19b specifically binds to Sox19b targets (ChIP experiment, Fig 1e), (ii) we do not see a developmental phenotype with ectopically expressed mEos2-Sox19b before sphere stage and (iii) we do not observe a sign of binding saturation are sufficient control to assume that mEos2-Sox19b approximates endogenous Sox19b binding within the limitations imposed by ectopic expression and an added fluorescent tag.

We now mention more clearly the limitation in the main text:

Overall, these experiments indicate that chromatin-binding of mEos2-TF approximates binding of endogenous TF within the limitations imposed by ectopic expression and an added fluorescent tag.

Reviewers' Comments:

Reviewer #1:

Remarks to the Author:

The authors have satisfactorily addressed my remaining comments and criticism.

Reviewer #3:

Remarks to the Author:

The manuscript has improved substantially from the original submission. As also noted by the other reviewers, the original manuscript contained many unsubstantiated claims and the present version instead is generally careful and restricts itself mostly to claims that are supported by the data. But it has been quite frustrating because the authors keep on overclaiming and pushing things even during the review process. As noted by reviewer 1, the authors previously claimed in their revision that the TBP overexpression was marginal, but it is actually 3.8-fold. Clearly a ~4-fold overexpression is not "marginal" and being told things that are not accurate over multiple cycles of review is quite frustrating, even if the authors eventually take out the overclaims.

As the authors have reported the first single-molecule imaging in live zebrafish embryos and new analysis techniques (ITM), the paper is exciting and worthy of publication. But how much time could all of us not have saved if the authors could just stick to the data instead of twisting things and overclaiming?

Anyway, in the interest of not wasting additional time, we will support publication once the following 4 items have been fully solved (no experiments required; should be possible to do this in a day or two):

1. Supplement Section 4.2.3 claims that the bound fraction is independent of TF concentration. This is not universally true (see below). In fact, the authors can only arrive at this result using an approximation (equation [30]). Thus, the authors should change the title of this section and clearly state that this is only true in a limited range and not universally true.
2. Authors should mention in the main text that saturation effects have been observed upon protein overexpression and cite the 2 examples mentioned below. It is great that the authors have evidence that TBP and Sox19 may not show saturation, but they should still honestly acknowledge that this is a known issue with overexpression and cite the 2 papers
3. The editor's email (Dr. Mieck) says that we as reviewers should only recommend publication if the raw data has been provided. Apologies if we have missed it, but we could not find the raw SPT data. It is probably overkill to publish all the raw movies, but the authors must at a minimum publish ALL the raw SPT trajectories for TBP and Sox19. E.g. DataDryad or another public repository can host these.
4. The authors must publish their code. Obviously, it is fine to leave out small things. But the 2 major pieces of code: ITM analysis (Figure 4) and time-lapse analysis to get residence times (Figure 3), must be published. We cannot support publication otherwise. The authors should put the code on a public repository like GitHub (or another one of their choosing). The authors should include a small tutorial on how to run the code as well as an example SPT dataset on which to run the code (data from this paper). This way, we can understand how the code works and how the figures are made. We note that it would have been quite helpful if the reviewers had done this from the beginning such that we reviewers could actually have assessed the analysis during the process. But at the very least, the ITM and time-lapse analysis code must be on a public repository before publication.

Response to reviewer responses:

The authors claim in their response to our review that "Instead, the true bound fraction is given by $(k_{on}[DNA])/(k_{off}+k_{on}[DNA])$ (see our extended derivation in the supporting info), which is

independent of TF concentration.”. This is obviously not universally true and it is unclear why the authors keep claiming this. It is true that under certain assumptions, this can be approximately correct, but that is very different from it being universally true.

When we referred to k_{ON} in the original response it was the pseudo-rate. In any case, authors and we agree that the fraction bound is given by:

$$f_{Bound} = (k_{ON}[DNA] / (k_{OFF} + k_{ON}[DNA]))$$

According to equation 30 in supplement page 19, $[DNA] = [DNA_0] - [DNA-TF]$

According to equation 29 in supplement page 18: $[DNA-TF] = [DNA]*[TF]/K$

Putting equation 29 and 30 together we get: $[DNA] = [DNA_0] - [DNA]*[TF]/K$

Rearranging: $[DNA] = [DNA_0]*K / (K+[TF])$

Now we put this into the expression for f_{Bound} :

$$f_{Bound} = (k_{ON}*[DNA_0]*K / (K+[TF])) / (k_{OFF} + k_{ON}*[DNA_0]*K / (K+[TF]))$$

In other words, f_{Bound} depends on $[TF]$. Authors assume that $[DNA_0] \gg [DNA-TF]$ in equation 30. This may be approximately correct in some cases, but it is obviously not universally true, and the authors make it sound like it is universally true.

Beyond the math: take a simple example. A cell has 10 sites for a TF and nothing more. As TF concentration increases, eventually all 10 sites are occupied. At this point $[DNA]=10$ and $[DNA-TF]=10$. Further increasing the TF concentration cannot increase the bound fraction because the cell has run out of sites. In this thought example, the number of bound TFs will saturate out at 10 and if we were to increase the TF concentration to the limit of infinity, the f_{Bound} would approach 0. Surely, the authors do not deny this?

The fact that the authors see a linear relationship in Figure 4c does suggest that saturation is likely not to be an issue in this specific case, but the authors clearly misunderstood our comments.

There are several examples in the literature of f_{Bound} decreasing with $[TF]$ concentration. The Agarwal paper shows defects upon TF overexpression, but that could be for any number of reasons. We clearly cited an example of “apparent saturation”

(<https://elifesciences.org/articles/25776>), but the authors seemed to ignore it. We have now done the work of digging through the figures for the authors. The relevant figures are Figure2-figureSupplement2B and Figure2-figureSupplement2D.

It is important not to confuse saturation of binding sites with physiological defects. They are likely to be related in some cases, but they are not the same. And you can easily imagine having saturation of binding sites without observable physiological defects (it will depend on the biology).

It is totally fair for the authors to argue that saturation of binding sites is unlikely to occur in their case, but they should acknowledge in the main text that it has been reported to occur. And they should cite Zhong Cell 2012 and (<https://elifesciences.org/articles/25776>) as published examples where saturation appears to occur.

REVIEWERS' COMMENTS:

Reviewer #3 (Remarks to the Author):

1. Supplement Section 4.2.3 claims that the bound fraction is independent of TF concentration. This is not universally true (see below). In fact, the authors can only arrive at this result using an approximation (equation [30]). Thus, the authors should change the title of this section and clearly state that this is only true in a limited range and not universally true.

We agree with the reviewer that the bound fraction is not universally independent of TF concentration. We therefore restricted our considerations to the explicit case of our measurements and already clearly indicated this (first sentence of Section 4.2.3: ...we do not observe saturation. We therefore can assume ...; Rewriting the law of mass action with this approximation...) This directly implies that our conclusions are only valid under this assumption and does not imply they are universally true. Nevertheless, to avoid misunderstanding and to make this even clearer, we changed the title of this section to: **4.3 The bound fraction of TFs**

We now again repeat the assumption directly after our conclusion:
As it can be seen from the right hand side of the equation above, Q is independent of the concentration of TF. **This is true as long as saturation does not occur but not universally true.**

Thus, we indeed find that $Q' = Q$ and Q' is independent of the concentration of TF. **This is true as long as saturation does not occur but not universally true.**

2. Authors should mention in the main text that saturation effects have been observed upon protein overexpression and cite the 2 examples mentioned below. It is great that the authors have evidence that TBP and Sox19 may not show saturation, but they should still honestly acknowledge that this is a known issue with overexpression and cite the 2 papers

We now cite the papers the reviewer is asking for:
... whether effects of over-expression, which depend on the factor of interest^{44,50-52}, ...

3. The editor's email says that we as reviewers should only recommend publication if the raw data has been provided. Apologies if we have missed it, but we could not find the raw SPT data. It is probably overkill to publish all the raw movies, but the authors must at a minimum publish ALL the raw SPT trajectories for TBP and Sox19. E.g. DataDryad or another public repository can host these.

The data archiving policy of Nature Communications changed during the review process. Initially, we did not store our data on a public repository.
We now deposited all raw SPT data for TBP and Sox19b in the online repository Dryad Data Repository. Data will be accessible upon publication of the manuscript.

4. The authors must publish their code. Obviously, it is fine to leave out small things. But the 2 major pieces of code: ITM analysis (Figure 4) and time-lapse analysis to get residence times (Figure 3), must be published. We cannot support publication otherwise. The authors should put the code on a public repository like GitHub (or another one of their choosing). The authors should include a small tutorial on how to run the code as well as an example SPT dataset on which to run the code (data from this paper). This way, we can understand how the code works and how the figures are made. We note that it would have been quite helpful if the reviewers had done this from the beginning such that we reviewers could actually have assessed the analysis during the process. But at the very least, the ITM and time-lapse analysis code must be on a public repository before publication.

The software policy of Nature Communications changed during the review process. Initially, we did not

store our code on a public repository.

We now published the code for time-lapse analysis and ITM analysis in the online repository Dryad Data Repository, together with the raw data and a step-by-step manual. Code and data will be accessible upon publication of the manuscript.

To make data and code more readily accessible to readers, we have reorganized and improved the storage structure of ITM data. The code accesses this new structure. In the course of this reorganization, we have re-analyzed parts of the data. This led to changes in molecule numbers due to the manual identification of nuclear area (highlighted in red in the manuscript). In addition, in the calculation of the stable bound proportion of chromatin-bound molecules, we introduced the criterion that at least two molecules have to be bound in a data set to enter analysis (Figure 5b). Moreover, a previously wrong correction factor for the bound fraction of Sox19b was corrected (Figure 5a). None of these changes altered our conclusions.

Additionally, the manuscript contains text changes to satisfy editorial style requirements.